# Universal Differential Equations for Stable Multi-Step Volatility Time Series Forecasting

**Prasanna Devadiga**                                           *devadigaprasanna28@gmail.com*
*TIFIN*

**Kishan Gurumurthy**                                           *kishangurumurthy@gmail.com*
*TIFIN*

**Kshitij Mohan**                                           *kshitijmohan15@gmail.com*
*TIFIN*

**Reviewed on OpenReview:** *https://openreview.net/forum?id=uWGNexco2M*

## Abstract

Neural differential equations such as Neural Ordinary Differential Equations (Neural ODEs), Neural Controlled Differential Equations (Neural CDEs), and Universal Differential Equations (UDEs) model temporal evolution as a continuous-time flow rather than a fixed-step recurrence. Even for regularly sampled data, this formulation differs fundamentally from discrete-time architectures: it learns smooth vector fields governing instantaneous rates of change, reducing discretization bias and improving long-horizon stability. We present a systematic study of **Universal Differential Equations** for financial volatility forecasting, a domain characterized by regime shifts, heavy tails, and jump discontinuities. UDEs extend Neural ODEs by embedding mechanistic structure within learned dynamics, using neural networks to parameterize coefficients in partially known differential equations instead of learning the system purely from data. Our UDE variants incorporate volatility's empirical regularities while retaining neural flexibility for regime adaptation. Our formulation approximates the aggregate impact of jumps through smooth, continuous dynamics rather than explicit stochastic arrivals, retaining tractability and interpretability. Across market regimes, they outperform both continuous-time baselines and discrete-time models, achieving higher accuracy and greater long-horizon stability while remaining interpretable. These results suggest that UDEs grounded in mechanistic structure and neural flexibility offer a principled route to stable, interpretable multi-step forecasting in nonstationary domains.

## 1 Introduction

Time series forecasting is fundamentally challenging due to complex temporal dependencies that make accurate prediction difficult (25; 19). These challenges are particularly acute in domains where the data-generating process itself is unstable and where small modeling errors can have cascading consequences.

**We focus on financial volatility as our evaluation domain:** Volatility serves as a demanding testbed for three reasons. First, it is practically important: volatility quantifies market risk and is fundamental to portfolio construction and systemic risk management (3; 11). Second, open data (from *yfinance*) enables reproducible research without computational burden. Third, volatility exhibits extreme versions of forecasting challenges through well-documented "stylized facts" (6): *clustering* (high-volatility periods persist), *jump discontinuities* (sudden spikes from large price movements), and *heavy-tailed distributions* (extreme events occur more frequently than normal distributions predict). These properties make even single-step prediction difficult, as models trained during calm markets fail catastrophically under regime shifts. While volatility dynamics are often modeled with **stochastic differential equations (SDEs)** incorporating explicit Poisson jump terms, such formulations introduce heavy computational and identifiability burdens. In this work,

we instead approximate jump effects through a smooth deterministic modulation within a continuous-time ODE framework, preserving differentiability and solver stability while retaining the key phenomenological behavior of sudden volatility bursts.

**Multi-step forecasting amplifies these challenges:** Errors compound with horizon, subtle misspecifications become catastrophic failures, and models must maintain stability while recursively feeding their own predictions forward. For volatility, clustering means one error during high-volatility regimes can destroy entire forecast trajectories—if the model underestimates at step $t$, it continues underestimating at $t+1, t+2, \ldots$ as clustering persists. Unlike smoother time series where multi-step errors degrade gradually, volatility forecasts collapse abruptly at longer horizons.

**Current approaches fall short in different ways:** This motivates hybrid approaches combining neural flexibility with mechanistic stability. Fully data-driven architectures (LSTMs, GRUs, Transformers) learn discrete-time transitions as black-box functions, requiring massive datasets and often failing at long horizons (25; 19). Neural ODEs improve inductive bias through continuous-time dynamics but still learn unconstrained vector fields from data (5). Mechanistic models (GARCH, Heston SDEs) encode domain knowledge but cannot adapt beyond fixed parametric forms (1; 3) and are computationally expensive to train when extended to full stochastic (SDE or jump-diffusion) formulations. UDEs bridge this gap by embedding mechanistic structure while allowing neural modulation of key coefficients.

**Universal Differential Equations bridge this gap:** An Ordinary Differential Equation (ODE) specifies a fully known righthand side, $\dot{x} = f(x, t; \vartheta)$; a Universal Differential Equation (UDE) augments a mechanistic ODE with a learned component, $\dot{x} = f_{\text{known}}(x, t; \vartheta) + g_{\text{NN}}(x, t; \phi)$, or equivalently learns state-dependent coefficients while retaining the governing form. In this paper, we adopt the modulation view: small MLPs modulate the mean reversion coefficients under positivity/bound constraints to preserve dissipativity and interpretability, in contrast to a vanilla neural ODE that learns an unconstrained vector field.

**UDEs for forecasting: A different problem class:** While neural ODEs and SDEs have been applied to various forecasting tasks, UDEs were introduced for system identification and scientific discovery, where models fit to complete observed trajectories with ground-truth feedback at each step (21). Multi-step forecasting presents fundamentally different challenges: models must maintain stability under *autoregressive rollout*, recursively feeding their own predictions forward without ground-truth correction. This autoregressive structure causes errors to compound over time. A small bias at horizon $h = 1$ becomes catastrophic degradation at $h = 20$ as the model repeatedly consumes its own outputs. To our knowledge, we provide the first systematic study demonstrating that UDEs, when properly constrained through compositional stacking of domain structure, remain stable under this autoregressive stress test, where unconstrained neural ODEs collapse (Section 2).

**Why not SDEs or Neural SDEs?** Stochastic differential equations provide a principled way to represent uncertainty, but neural or jump-augmented SDEs are computationally expensive and often numerically unstable. Training requires stochastic integration and Monte-Carlo gradient estimation at every step, leading to high variance and long runtimes. Moreover, diffusion and jump parameters are difficult to identify jointly from historical volatility alone without exogenous information (options data, macro factors). To maintain tractability and interpretability, we focus on deterministic UDEs that embed financial structure while remaining computationally efficient and stable under long rollouts.

**Our approach: Compositional UDEs for volatility:** We use neural networks to modulate mechanistic coefficients within a governing differential equation rather than learning dynamics from scratch. For volatility forecasting specifically, we incorporate mean-reversion dynamics with task-specific coefficients: mean-reversion rates (speed of return to equilibrium) and long-term volatility levels. This preserves the stabilizing inductive bias of the mechanistic form while allowing adaptation to regime-dependent dynamics. The result is a model that maintains interpretability each learned coefficient corresponds to a meaningful market quantity while achieving strong multi-step stability. Our framework leverages volatility's stylized facts as domain knowledge while allowing neural components to learn continuous, state-dependent parameter variations. The mechanistic structure constrains dynamics to remain mean-reverting by construction, while neural modulation enables smooth adaptation to evolving market conditions within that stable template.

**Model hierarchy and terminology:** Our experimental framework spans a spectrum from pure data-driven to pure mechanistic models. At one extreme, **Vanilla Neural ODE** learns dynamics entirely from data with no structural priors. At the other, **ODE-MR** is a pure mean-reverting process with fixed parameters. Between these extremes, our UDE variants progressively incorporate stylized facts: **UDE-MR** adds neural modulation of mean-reversion coefficients; **UDE-MRVC** further adds volatility clustering; and **UDE-MRVCJ** additionally incorporates jump dynamics through a smooth deterministic modulation term that approximates average jump effects within a continuous-time formulation. All UDE variants enforce soft constraints, such as nonnegative mean-reversion rates, bounded long-run levels, and structured clustering terms to ensure neural components respect rather than overturn the mechanistic structure.

**Our contributions are:**

1. **Application of UDEs to autoregressive multi-step forecasting:** Unlike previous UDE applications in system identification where models fit complete observed trajectories, multi-step forecasting requires stability under repeated self-feeding of predictions without ground truth correction. We demonstrate that structurally constrained UDEs maintain performance under this autoregressive stress test, using rigorous rolling walk-forward validation across asset classes.

2. **Parameter efficiency through structural inductive biases:** UDEs achieve competitive accuracy with significantly fewer parameters than foundation models, demonstrating that embedding domain structure can replace massive scale for forecasting tasks with partial mechanistic knowledge.

3. **Interpretable regime adaptation:** Unlike black-box foundation models, the hybrid architecture maintains interpretability through its mechanistic structure. Learned parameters correspond to economically meaningful quantities (mean-reversion rates, volatility clustering strength, jump intensities) that can be tracked across time, enabling post-hoc analysis of how market dynamics evolve across regimes (Figure 2).

4. **Long-horizon stability:** UDEs remain relatively stable across longer horizons, unlike purely neural and foundation models, which degrade at a higher rate.

All results are reported in trading days, where 1, 5, 10, 20 steps correspond approximately to daily, weekly, bi-weekly and monthly time-periods periods for most assets.

## 2 Related Work

**Multi-step Time Series Forecasting:** Multi-step forecasting presents unique challenges due to error accumulation and distribution shift, with approaches broadly categorized into direct, recursive, and multi-output strategies (23). Deep learning architectures have significantly advanced the field, including LSTNet (17) which combined CNNs and RNNs for temporal patterns, and transformer-based methods like Informer (28) and Autoformer (25) that capture long-term dependencies through specialized attention mechanisms. Recent state-of-the-art foundation models like TimesFM (8) leverage large-scale pretraining across diverse temporal domains to enable robust, zero-shot multi-step forecasting.

**Neural Differential Equations for Time Series:** Neural ordinary differential equations (Neural ODEs) (5) model temporal evolution as continuous dynamical systems, enabling natural interpolation and extrapolation. They have been applied to various forecasting domains including energy demand prediction (26), weather forecasting (24), and ecological population dynamics (2). Neural stochastic differential equations (Neural SDEs) extend this framework by incorporating stochastic terms for uncertainty quantification, finding applications in retail sales forecasting and financial modeling (9). Neural controlled differential equations (Neural CDEs) (16) have been evaluated on standard time series benchmarks, demonstrating advantages for irregularly-sampled data. However, these works primarily focus on system modeling or short-horizon prediction tasks, with evaluation protocols typically using limited validation strategies.

**Volatility Forecasting:** Financial volatility forecasting has evolved from parametric econometric models to flexible machine learning approaches. The GARCH family (3; 12) and stochastic volatility models (13) have

dominated academic finance, providing interpretable but potentially misspecified parametric forms. Recent machine learning approaches include neural networks for volatility prediction (27), deep learning models capturing complex patterns (4), and hybrid methods that combine GARCH with neural networks as feature extractors (18).

**Universal Differential Equations:** The Universal Differential Equations paradigm represents an advancement in scientific machine learning, structurally integrating mechanistic models with data-driven components. Introduced by Rackauckas et al. (21), UDEs parameterize unknown dynamical terms using neural networks, enabling simultaneous leverage of domain knowledge and data-driven flexibility. This extends several key lineages: Neural ODEs (5), Physics-Informed Neural Networks (22), and symbolic regression for interpretable dynamics (7). While previous UDE applications have focused on physical and biological systems, our work explores their value for financial forecasting, modeling volatility dynamics as partially-specified differential equations with learned market microstructure effects.

## 3 Method

We formulate financial volatility forecasting as a multi-step prediction problem. Let $\{x\}_{t=0}^T$ denote the sequence of log-volatilities, where $x_t = \ln(\sigma_t)$ and $\sigma_t$ represents the latent daily volatility. The objective is to predict future log-volatilities $\hat{x}_{t+h}$ at fixed horizon $h$, leveraging past observed values within a look-back window of length $K$.

Log-volatility is the standard target for volatility modeling. It acts as a variance-stabilizing transformation: the original volatility process $\sigma_t$ exhibits heteroskedasticity, while $x_t = \ln(\sigma_t)$ converts multiplicative dynamics into additive noise with more stable variance, simplifying modeling assumptions across methods.

Volatility is estimated using the Garman-Klass (GK) range-based estimator on daily prices. For each day $t$, the GK realized variance is

$$\sigma_{\text{GK},t}^2 = \tfrac{1}{2}\left(\ln \tfrac{H_t}{L_t}\right)^2 - \left(2\ln 2 - 1\right)\left(\ln \tfrac{C_t}{O_t}\right)^2 \tag{1}$$

where $O_t, H_t, L_t, C_t$ are the open, high, low, and close. We take $\sigma_t = \sqrt{\sigma_{\text{GK},t}^2}$ and use $x_t = \ln(\sigma_t)$ for modeling. The GK estimator is approximately 7.4 times (10) more efficient than the close-to-close squared returns volatility estimator for daily data while remaining robust to drift.

### 3.1 Pure Physics Model: ODE with Mean Reversion

Considering the log-volatility $x_t \in \mathbb{R}$, the pure physics model posits the continuous-time dynamics governed by the Ordinary Differential Equation:

$$d(x)/d(t) = \kappa(\theta - x) \tag{2}$$

with parameters $\kappa > 0$ representing the mean reversion speed and $\theta \in \mathbb{R}$ the long-term equilibrium. This linear vector field is globally Lipschitz continuous with Lipschitz constant $\kappa$, which by the Picard–Lindelöf theorem guarantees existence and uniqueness of solutions for all initial conditions. The closed-form solution

$$x(t) = \theta + (x_0 - \theta)e^{-\kappa t} \tag{3}$$

demonstrates exponential convergence to $\theta$, confirming global asymptotic stability. The Jacobian matrix of the system linearised about the equilibrium point $x^* = \theta$ is $J = -\kappa < 0$, which ensures perturbations decay exponentially, embodying the mean reversion property fundamental to modeling volatility dynamics.

### 3.2 Pure Neural ODE Model: Data-Driven Dynamics

In contrast to the purely parametric physics model, the pure neural ODE represents the log-volatility evolution via a learnable continuous vector field parameterized by a neural network $NN_\phi : \mathbb{R} \to \mathbb{R}$,

$$d(x)/d(t) = NN_\phi(x), \tag{4}$$

where $\phi$ denotes network parameters. Leveraging the Universal Approximation Theorem, this neural network, with sufficient depth and non-linear smooth activations such as GELU, can approximate any continuous vector field on a compact subset of $\mathbb{R}$ arbitrarily well. The local Lipschitz continuity of $NN_\phi$ ensures, via classical ODE theory, local existence and uniqueness of solutions. However, neural vector fields impose no inherent structural guarantees like mean reversion or boundedness, potentially leading to unstable trajectories or divergence, which may affect forecasting performance and interpretability.

### 3.3 Universal Differential Equation Model: Structural Coupling via Neural Modulation

The universal differential equation framework allows us to integrate prior domain knowledge by coupling neural modulation with mean reversion, one of the strongest studied effects in volatility:

$$\frac{dx}{dt} = \kappa_{\text{eff}}(x) \left( \theta_{\text{eff}}(x) - x \right) \tag{5}$$

Rather than learning the dynamics function entirely from data (as in Vanilla Neural ODE) or fixing parameters rigidly (as in pure ODE), we use small neural networks to modulate the mechanistic coefficients in a state-dependent manner. Specifically, the mean-reversion rate and long-term level are defined as:

$$\kappa_{\text{eff}}(x) = \underbrace{\text{softplus}(\kappa) + \varepsilon}_{\kappa_+} \cdot \exp\big(s \tanh(NN_1(x))\big), \quad \theta_{\text{eff}}(x) = \theta + b \tanh(NN_2(x)) \tag{6}$$

where $NN_1$ and $NN_2$ are small MLPs (e.g., 3 hidden layers with 64 units each), $s > 0$ and $b > 0$ control the modulation strength, and $\varepsilon > 0$ is a small constant. This multiplicative structure ensures $\kappa_{\text{eff}}(x) > 0$ (preserving dissipativity) while keeping $\theta_{\text{eff}}(x)$ bounded, thereby maintaining mean-reverting stability. Critically, the neural component modulates coefficients within the mechanistic structure rather than adding a separate unconstrained drift term, which would allow the network to learn dynamics that overturn the mean-reverting property.

**Incremental model composition.** We first enforce physics-parameter constraints ($\kappa_+ > 0$ via softplus and bounded $\theta_{\text{eff}}$ shifts), which forces a mean-reverting representation and prevents runaway dynamics. Building on this base, we progressively incorporate additional stylized facts through compositional stacking: volatility clustering (via lagged squared returns) and jump processes (via learned jump rate and magnitude). This progressive design enables systematic ablation of each component's contribution to forecasting performance. Table 1 presents the model variants evaluated in our experiments.

**Integrator choice:** Because the volatility data are uniformly sampled at daily frequency, we employ a fixed-step fourth-order Runge–Kutta (RK4) integrator with step size $h = 1$ day for all models. This choice avoids unnecessary adaptive stepping overhead while maintaining high-order accuracy and deterministic reproducibility across thousands of rolling windows. Although adaptive solvers are beneficial for irregularly sampled data, our setting benefits more from the continuous-time formulation's structural properties: learning a smooth vector field that governs instantaneous rates of change rather than discrete one-step transitions. This continuous representation reduces discretization bias and promotes trajectory-level stability during multi-step rollouts, even under regular sampling.

Table 1: UDE model variants with progressive incorporation of stylized facts. Ablation studies (Section 4.3) show monotonic improvements from compositional stacking.

| Variant | Dynamics Equation |
|---|---|
| **UDE-Mean Reversion** | $\dfrac{dx}{dt} = \kappa_{\text{eff}}(x)\,(\theta_{\text{eff}}(x) - x)$ |
| **UDE-Volatility Clustering** | $\dfrac{dx}{dt} = \kappa_{\text{eff}}(x)\,(\theta_{\text{eff}}(x) - x) + \gamma_+\,r_{t-1}^2$ |
| | where $\gamma_+ = \text{softplus}(\gamma) + \varepsilon$ |
| **UDE-Jumps** | $\dfrac{dx}{dt} = \kappa_{\text{eff}}(x)\,(\theta_{\text{eff}}(x) - x) + \lambda(x)\,J(x)$ |
| | where $\lambda(x) = \text{softplus}(NN_\lambda(x)) + \varepsilon,\ J(x) = \tanh(NN_J(x))$ |
| **UDE-MRVCJ** | $\dfrac{dx}{dt} = \kappa_{\text{eff}}(x)\,(\theta_{\text{eff}}(x) - x) + \gamma_+\,r_{t-1}^2 + \lambda(x)\,J(x)$ |

# 4 Experiments

We evaluate our UDE-based volatility forecasting framework using rigorous rolling walk-forward validation across diverse financial assets. Our experimental design tests model robustness across varying market dynamics, volatility regimes, and stress periods spanning nearly two decades.

## 4.1 Datasets and Evaluation Protocol

**Asset Classes:** We evaluate on 10 financial indices spanning major global markets and asset classes: developed and emerging market equities, government bonds, small/large cap indices, and cryptocurrency. This diverse collection exhibits strong temporal dependencies, conditional heteroskedasticity, and stress periods, including the 2008 financial crisis, 2020 COVID crash, and 2022 market turbulence. Daily prices are obtained from Yahoo Finance, and volatility is estimated using the Garman–Klass range-based estimator. We model log-volatility $x_t = \ln(\sigma_t)$ as described in Section 3. Complete details and tickers are provided in Appendix A.3.

**Rolling walk-forward validation:** We use strictly non-overlapping rolling windows to ensure realistic out-of-sample evaluation. Each window consists of sequential train, validation, and test splits with no data leakage. Windows are rolled forward through the entire sample period (Aug 2005 - Aug 2025), naturally traversing multiple market regimes and crisis periods. This protocol prevents look-ahead bias and provides conservative performance estimates. Full details on window sizes, stride length, and split configuration are provided in Appendix A.4.

**Baselines:** We compare against statistical models (Random Walk, EWMA, GARCH-family), deep learning approaches (LSTM, GRU), foundation models (TimesFM zero-shot and few-shot), pure mechanistic baselines (ODE-MR), and pure neural ODE baselines (Vanilla Neural ODE). This comprehensive set isolates the contribution of hybrid structural constraints.

## 4.2 Results

We present the performance evaluation of our UDE-based architecture across multiple forecasting horizons, comparing against traditional econometric models, deep learning baselines, and foundation models. Table 2 shows aggregated results across horizons 1, 5, 10, and 20 steps ahead.

Table 2: Normalized RMSE (mean $\pm$ 95% CI). Normalized by standard deviation of observed volatility. Scale-independent metric, lower is better.

| Model | Type | NRMSE $h = 1$ | NRMSE $h = 5$ | NRMSE $h = 10$ | NRMSE $h = 20$ |
|---|---|---|---|---|---|
| Random Walk | Baseline | $0.770 \pm 0.063$ | $0.841 \pm 0.080$ | $0.882 \pm 0.084$ | $0.911 \pm 0.087$ |
| EWMA | Statistical | $1.381 \pm 0.220$ | $1.442 \pm 0.223$ | $1.498 \pm 0.232$ | $1.602 \pm 0.245$ |
| GARCH(1,1) | Econometric | $1.384 \pm 0.222$ | $1.494 \pm 0.234$ | $1.606 \pm 0.247$ | $1.852 \pm 0.277$ |
| FIGARCH(1,1) | Econometric | $1.430 \pm 0.235$ | $1.548 \pm 0.249$ | $1.677 \pm 0.267$ | $1.985 \pm 0.314$ |
| LSTM | Sequence | $0.811 \pm 0.052$ | $0.825 \pm 0.048$ | $0.855 \pm 0.049$ | $0.965 \pm 0.047$ |
| GRU | Sequence | $0.803 \pm 0.048$ | $0.817 \pm 0.046$ | $0.818 \pm 0.048$ | $0.906 \pm 0.046$ |
| MertonJD | Jump Diffusion | $0.834 \pm 0.083$ | $0.892 \pm 0.115$ | $0.929 \pm 0.165$ | $0.964 \pm 0.196$ |
| TimesFM | Transformer | $0.704 \pm 0.055$ | $0.782 \pm 0.051$ | $0.794 \pm 0.053$ | $0.819 \pm 0.050$ |
| TimesFM few-shot | Transformer | $0.701 \pm 0.054$ | $0.780 \pm 0.050$ | $0.791 \pm 0.052$ | $0.814 \pm 0.049$ |
| Vanilla Neural ODE | Neural ODE | $0.735 \pm 0.056$ | $0.812 \pm 0.073$ | $0.949 \pm 0.228$ | $0.998 \pm 0.090$ |
| ODE-MR | Physics | $0.758 \pm 0.080$ | $0.985 \pm 0.070$ | $0.997 \pm 0.069$ | $1.020 \pm 0.085$ |
| UDE-MR | UDE | $0.662 \pm 0.056$ | $0.740 \pm 0.050$ | $0.753 \pm 0.050$ | $0.780 \pm 0.050$ |
| UDE-MRVC | UDE | $0.660 \pm 0.056$ | $0.738 \pm 0.051$ | $0.749 \pm 0.052$ | $0.774 \pm 0.052$ |
| UDE-MRVCJ | UDE | $\mathbf{0.658} \pm 0.052$ | $\mathbf{0.735} \pm 0.048$ | $\mathbf{0.743} \pm 0.049$ | $\mathbf{0.759} \pm 0.048$ |

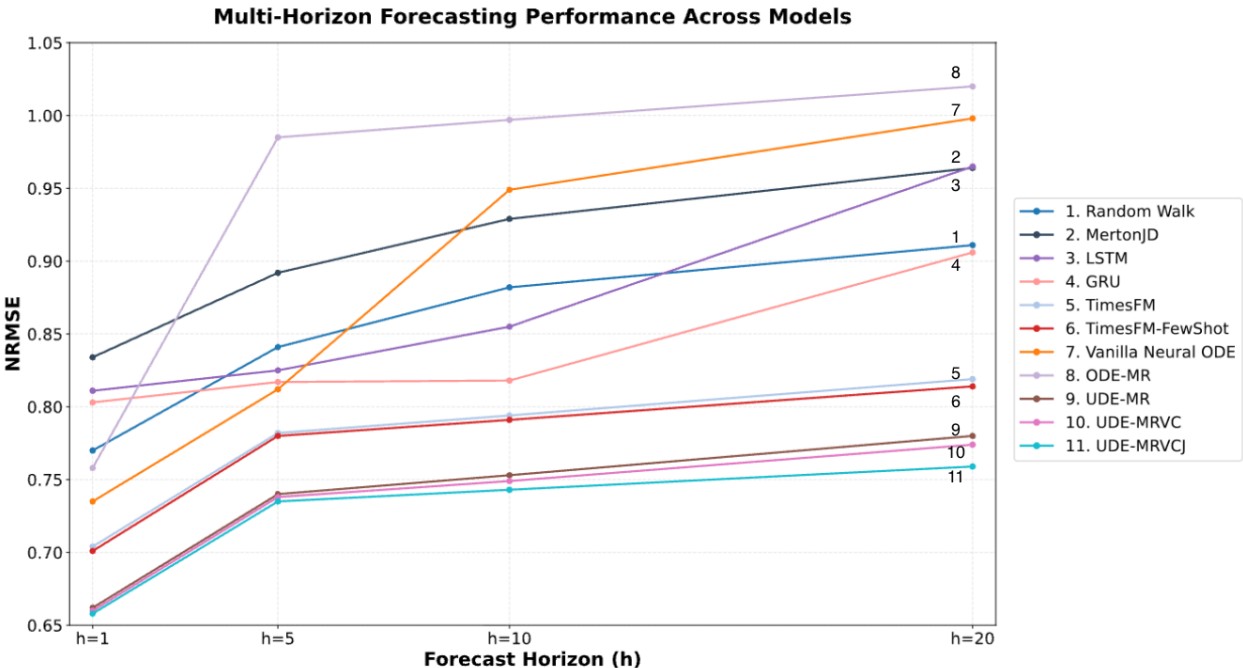

Figure 1: Multi-horizon forecasting accuracy (NRMSE; lower is better) at horizons $h \in \{1, 5, 10, 20\}$ for baseline (Random Walk), sequence models (LSTM, GRU), transformer models (TimesFM, TimesFM-FewShot), and Neural-ODE/UDE variants. The UDE-MRVCJ model (incorporating mean reversion, volatility clustering, and jumps) achieves the best performance across all horizons. Curves show mean results over rolling evaluation windows; lower lines indicate more accurate forecasts.

We evaluate significance using the Diebold–Mariano (DM) test across rolling windows for each asset. Because the selected assets span distinct classes (equity indices, emerging-market indices, and crypto) with low cross-

Table 3: Diebold-Mariano significance across assets (combined p-values, Fisher). Lower p indicates model 2 significantly outperforms model 1.

| Pair | $h = 1$ | $h = 5$ | $h = 10$ | $h = 20$ |
|---|---|---|---|---|
| Vanilla Neural ODE $\rightarrow$ UDE-MR | 3.1e-4 | 1.8e-5 | 2.2e-6 | 8.5e-7 |
| UDE-MR $\rightarrow$ UDE-MRVC | 0.0170 | 0.0857 | 0.0709 | 0.0386 |
| UDE-MRVC $\rightarrow$ UDE-MRVCJ | 0.0412 | 0.0013 | 0.0034 | 0.0189 |

correlations in realized volatility, we treat test outcomes as approximately independent. Combined p-values are therefore computed using Fisher's method.

### 4.2.1   Performance Analysis

**UDEs combine accuracy and stability.**   UDE variants slightly outperform the foundation model (TimesFM: 0.70; UDE-MRVCJ: 0.66) at short horizons. Across longer horizons, UDEs remain stable (0.66 $\rightarrow$ 0.76 from $h = 1$ to $h = 20$), whereas purely neural models degrade (Vanilla Neural ODE: 0.74 $\rightarrow$ 1.00) and rigid mechanistic models stay suboptimal (ODE-MR: 0.76 $\rightarrow$ 1.02).

**Compositional stacking yields monotonic improvements.**   Table 3 shows statistically significant improvements at each composition step. Adding mean-reversion modulation to Vanilla Neural ODE provides the largest gain (DM p < 3.1e-4 at $h = 1$, p < 8.5e-7 at $h = 20$). Volatility clustering (UDE-MRVC) and jumps (UDE-MRVCJ) provide incremental but consistent improvements, particularly at longer horizons where structural constraints prevent catastrophic degradation.

### 4.2.2   Computational Efficiency

Table 4: The efficiency table highlight that UDE variants are *orders of magnitude* smaller and have a smaller footprint than large neural and foundation baselines while preserving long-horizon stability.

| Model | Params | Memory (MB) | Train (s) | Infer (s) | Total (s) | Epoch (s) |
|---|---|---|---|---|---|---|
| Random Walk | 0 | 0.00 | N/A | 0.00 | 0.00 | 0.000 |
| EWMA | 1 | 0.05 | N/A | 0.00 | 0.00 | 0.000 |
| GARCH(1,1) | 4 | 0.10 | 0.36 | 0.00 | 0.36 | 0.000 |
| FIGARCH(1,1) | 5 | 0.10 | 1.11 | 0.00 | 1.11 | 0.000 |
| Merton JD | 8 | 0.15 | 0.95 | 0.02 | 0.97 | 0.002 |
| LSTM | 50.5K | 0.19 | 0.49 | 0.19 | 0.68 | 0.010 |
| GRU | 37.9K | 0.14 | 0.46 | 0.18 | 0.64 | 0.009 |
| ODE-MR | 2 | 0.00 | 0.50 | 0.02 | 0.52 | 0.005 |
| Vanilla Neural ODE | 4.4K | 0.02 | 1.02 | 0.05 | 1.07 | 0.010 |
| UDE-MR | 4.4K | 0.02 | 1.32 | 0.09 | 1.42 | 0.013 |
| UDE-MRVC | 4.4K | 0.02 | 1.46 | 0.10 | 1.56 | 0.014 |
| UDE-MRVCJ | 4.4K | 0.03 | 1.54 | 0.11 | 1.65 | 0.016 |
| TimesFM | 200.0M | 800.00 | N/A | 25.52 | 25.52 | 0.000 |
| TimesFM-FewShot | 200.0M | 800.00 | N/A | 25.25 | 25.25 | 0.000 |

### 4.3   Ablation Study

Table 5: Ablation study of model components for multi-step volatility forecasting. R² measures explained variance; sMAPE measures scale-independent forecast accuracy (lower is better).

| Model | Structure | $h = 1$ | | $h = 20$ | |
|---|---|---|---|---|---|
| | | **$\mathbf{R}^2$** | **sMAPE** | **$\mathbf{R}^2$** | **sMAPE** |
| ODE-MR | Pure Mechanistic | 0.421 | 7.91 | -0.198 | 11.58 |
| Vanilla Neural ODE | Pure Neural | 0.456 | 7.50 | -0.136 | 10.46 |
| UDE-MR | + Mean Reversion | 0.518 | 7.17 | 0.150 | 9.37 |
| UDE-MRVC | + Vol. Clustering | 0.522 | 7.14 | 0.151 | 9.36 |
| UDE-MRVCJ | + Jumps | 0.535 | 7.03 | 0.160 | 9.11 |

### 4.4   Interpretability via Parameter Evolution (S&P 500)

We track how the learned parameters of our UDE–MRVCJ evolve across rolling windows for the S&P 500. This reveals where the model adapts and where it remains invariant. First, the mean–reversion speed $\kappa(x)$ and the long–run level $\theta(x)$ co-move with regime shifts: $\kappa$ decreases during prolonged drawdowns (slower pullback toward equilibrium) and $\theta$ rises when background volatility is persistently elevated. Second, the jump mechanism is event–sensitive: the estimated jump *rate* $\lambda(x)$ exhibits pronounced spikes around market stress (e.g., macro announcements, crisis windows), while the jump *magnitude* concentrates within those episodes, indicating a parsimonious "rare event" channel rather than persistent overfitting. Third, the volatility–clustering strength ($\gamma$) varies smoothly and remains bounded, capturing short–horizon persistence without destabilizing the core dynamics. Finally, a stability summary and correlation heatmap show that core parameters ($\kappa, \theta$) remain stable across windows and are only weakly confounded with jump/clustering terms, supporting identifiability and interpretability. Together, these diagnostics let us explain forecast changes (e.g., "volatility increased because mean reversion slowed and jump intensity spiked") rather than treat predictions as opaque.

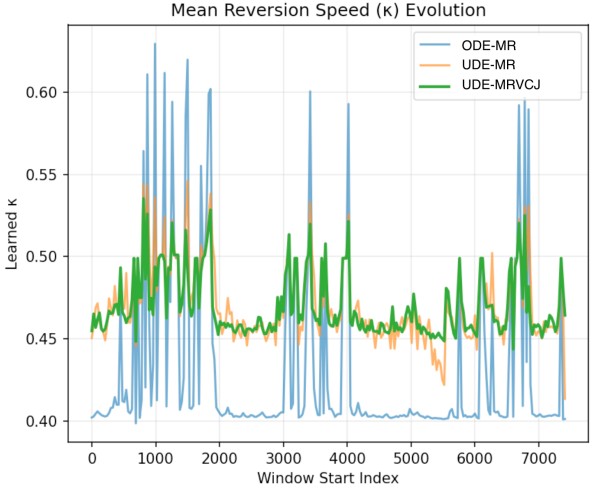

(a) Mean-reversion speed $\kappa(x)$: Controls how quickly volatility returns to equilibrium. Higher values during stress indicate faster normalization.

(b) Long–run level $\theta(x)$ across windows (S&P 500). Equilibrium volatility target. Increases during crises (elevated baseline risk), decreases during calm periods.

Figure 2: Learned mean-reversion dynamics across market regimes. Pure mechanistic baseline uses fixed parameters and oscillates aggressively in response to regime shifts, while UDE variants' neural components absorb these corrections, leading to more stable and meaningful parameters.

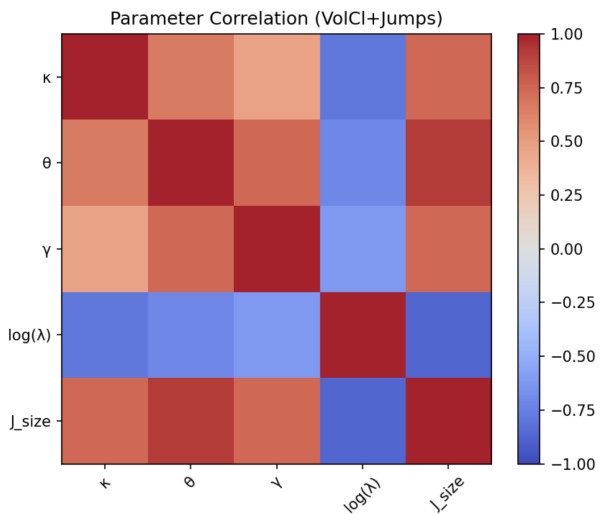

(a) Parameter correlation: weak confounding across channels (S&P 500).

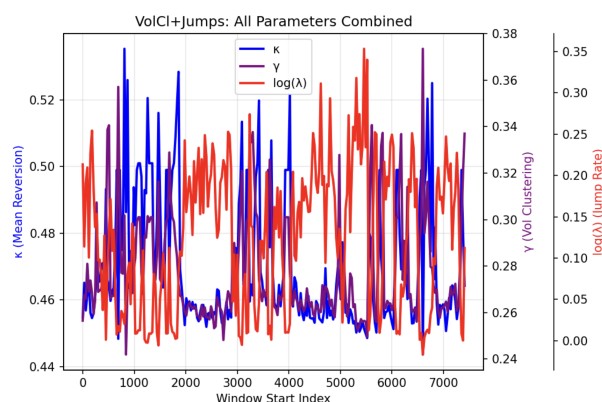

(b) Composite view: clustering + jumps alignment with stress (S&P 500).

Figure 3: Identifiability: mean–reversion parameters are largely orthogonal to jump/clustering terms.

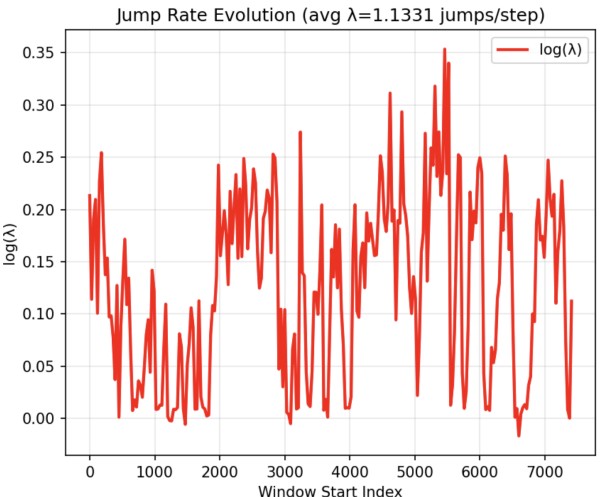

(a) Jump rate $lambda(x)$: Average jumps per step. Spikes during stress periods indicate the model learns when jump-like dynamics are necessary.

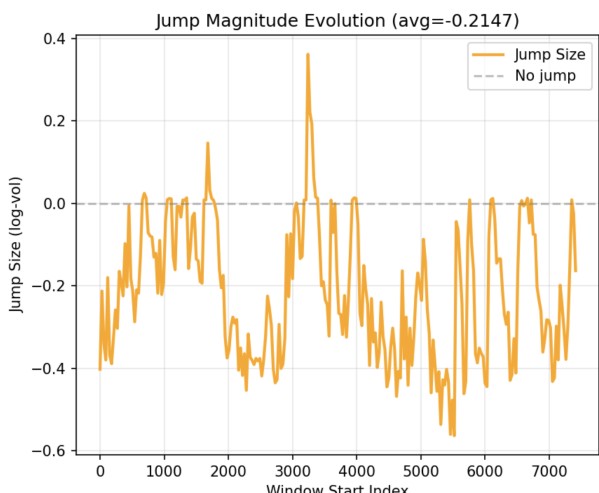

(b) Jump magnitude $J(x)$: Size of jump effect when activated. Concentrated during tail episodes, near-zero otherwise.

Figure 4: Jump dynamics activation during market stress (S&P 500). Our smooth, continuous jump formulation approximates the aggregate effect of discontinuous jumps through deterministic modulation.

## 5 Limitations and Future Work

**Point forecasts vs probabilistic predictions.** Our framework produces deterministic point forecasts rather than distributional predictions with uncertainty quantification. While sufficient for demonstrating Multi-step stability, risk management applications require value-at-risk estimation or full predictive distributions. Neural Stochastic Differential Equations (Neural SDEs) offer a natural extension by adding learned diffusion terms:

$$dx/dt = \kappa_{\text{eff}}(x)[\theta_{\text{eff}}(x) - x] + \gamma_{\text{eff}}(x)dW_t \tag{7}$$

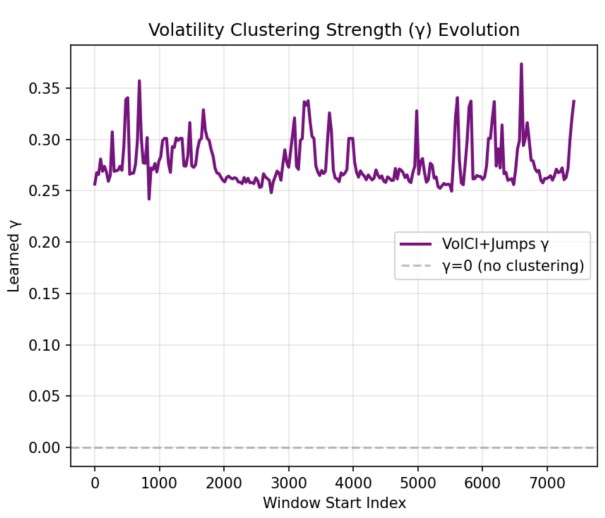

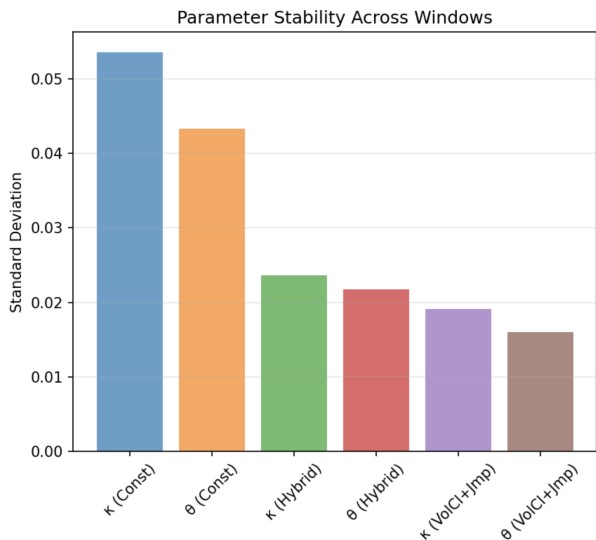

(a) Clustering strength $\gamma$: smooth, bounded variation (S&P 500).

(b) Stability summary: core parameters vary modestly. Consistent values across windows indicate the model learns general representations rather than overfitting.

Figure 5: Parameter stability and clustering dynamics (S&P 500). Volatility clustering strength varies smoothly without destabilizing core dynamics. Standard deviations across rolling windows confirm core mechanistic parameters remain stable while jump/clustering terms show comparable consistency.

where $\gamma_{\text{eff}}(x)$ models state-dependent volatility and $W_t$ represents Brownian motion. However, Neural SDEs remain notoriously difficult to train: adjoint methods require solving backward through stochastic processes with numerical instability (15). In contrast, the diffusion coefficient design requires careful constraints to prevent explosive trajectories (20). Recent advances in stable Neural SDE classes and signature-based training (14) suggest promising directions, but adapting these to compositional UDE frameworks remains open work.

**Limited exogenous features**: We use only lagged squared returns as exogenous input. Incorporating macroeconomic indicators, market microstructure features, or cross-asset correlations could improve regime detection and forecasting accuracy. In practice, longer-horizon volatility dynamics often depend on macroeconomic indicators, option-implied volatility, and cross-asset information. Integrating these features could enhance regime detection and long-range accuracy, though it risks diluting the structural clarity and parsimony that make UDEs interpretable. A key open question is how to incorporate exogenous signals without sacrificing mechanistic transparency.

**Domain generalization**: While financial volatility provides an excellent testbed due to documented stylized facts and accessible data, demonstrating the framework's effectiveness across diverse domains would strengthen claims of general methodological contribution. Each domain would require careful physics term design and validation against domain-specific baselines.

Despite these limitations, our core results remain robust: UDE variants with inductive biases achieve stable multi-step forecasting with dramatic parameter efficiency and interpretable dynamics. We explicitly note that institutional-grade data would be required for deployment in real-world trading or risk management systems.

## 6 Conclusion

We demonstrated that Universal Differential Equations (UDEs) provide an effective framework for multi-step time series forecasting, achieving superior accuracy and long-horizon stability across diverse financial

regimes, including crises and structural breaks. Our compositional UDE variants consistently outperform foundation, econometric, and deep learning baselines while using orders of magnitude fewer parameters.

UDEs and neural differential equations more broadly are particularly well suited to forecasting because they model temporal evolution as a **continuous flow** rather than a fixed-step recurrence. Even with regularly sampled data, this formulation enforces trajectory-level smoothness and reduces discretization bias, the main driver of error accumulation in recurrent architectures. By learning instantaneous rates of change instead of discrete transitions, UDEs maintain stable dynamics over extended horizons.

More fundamentally, UDEs offer a principled bridge between mechanistic and data-driven modeling. By embedding domain structure, mean reversion, volatility clustering, and jump effects within a differential equation while allowing neural modulation of key coefficients, they preserve **stabilizing inductive biases** that confine the learned dynamics to economically plausible regimes. Our compositional framework showed that each successive incorporation of domain-informed structure yields **monotonic, statistically significant gains**, demonstrating that carefully chosen inductive biases can substitute for brute-force scale.

This combination of continuous-time modeling and structured knowledge integration explains why compact UDE models can rival or surpass massive foundation models. Structural bias, when aligned with the data-generating process, not only enhances efficiency and interpretability but also delivers robust multi-step forecasts that remain coherent across regimes, an essential property for modeling inherently unstable systems like financial volatility.

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

## A   Appendix

### A.1   Lipschitz Continuity for UDE-MRVCJ

We consider the one-dimensional ODE on log-volatility

$$\frac{dx}{dt} = f(x; r_{t-1}^2) = \underbrace{\kappa_{\text{eff}}(x)\left(\theta_{\text{eff}}(x) - x\right)}_{\text{mean reversion}} + \underbrace{\gamma_+ r_{t-1}^2}_{\text{vol. clustering}} + \underbrace{\lambda(x)\, J(x)}_{\text{expected jumps}} \;,$$

where

$$\kappa_{\text{eff}}(x) = \kappa_0 \exp\!\big(s\, \tanh(k(x))\big), \qquad \theta_{\text{eff}}(x) = \theta + b\, \tanh(\vartheta(x)),$$

$$\lambda(x) = \text{softplus}(\ell(x)) + \varepsilon, \qquad J(x) = \tanh\big(j(x)\big).$$

Here $\kappa_0 = \text{softplus}(\kappa) + \varepsilon > 0$ is constant, $s > 0$ is the modulation scale, $b > 0$ is the $\theta$-shift bound , and $\varepsilon > 0$ is a fixed numerical slack. The maps $k, \vartheta, \ell, j : \mathbb{R} \to \mathbb{R}$ are the scalar outputs of two MLPs with GELU activations (one for $\{k, \vartheta\}$ and one for $\{\ell, j\}$). The clustering input $r_{t-1}^2$ is exogenous and treated as constant in $x$ at each step. We assume the MLPs are globally Lipschitz with constants $L_{\text{base}}$ (for $k, \vartheta$) and $L_{\text{jump}}$ (for $\ell, j$). For GELU networks this holds with

$$L_{\text{MLP}} \leq \left(\prod_i \|W_i\|_2\right) (1.129)^{\#\text{activations}}.$$

We write $K = [-M, M]$ for a compact domain of interest and use $\|\cdot\|$ for the Euclidean norm on $\mathbb{R}$.

**Auxiliary bounds:** (i) Since $|\tanh'| \leq 1$ and $(\text{softplus})' = \sigma \in (0, 1)$,

$$|(\tanh \circ g)'| \leq |g'|, \qquad |(\text{softplus} \circ g)'| \leq |g'|, \qquad \forall g : \mathbb{R} \to \mathbb{R}.$$

(ii) For $x \in K$, define

$$\kappa_{\max} := \kappa_0\, e^s, \qquad L_\kappa := \kappa_0\, e^s\, s\, L_{\text{base}}, \qquad L_\theta := b\, L_{\text{base}}.$$

Then $|\kappa_{\text{eff}}(x)| \leq \kappa_{\max}$ and $|\kappa'_{\text{eff}}(x)| \leq L_\kappa$; also $|\theta'_{\text{eff}}(x)| \leq L_\theta$ and $|\theta_{\text{eff}}(x)| \leq |\theta| + b$. (iii) For the jump factors, $|J(x)| \leq 1$ and

$$|\lambda'(x)| \leq L_{\text{jump}}, \qquad |J'(x)| \leq L_{\text{jump}}.$$

Let

$$\Lambda(M) := \sup_{x \in K} \lambda(x) \leq \log 2 + |\ell(0)| + L_{\text{jump}}\, M + \varepsilon$$

(using $\text{softplus}(t) \leq \log(1 + e^{|t|}) \leq \log 2 + |t|$ and $|\ell(x)| \leq |\ell(0)| + L_{\text{jump}}|x|$).

**Theorem A.1** (Local Lipschitz on compact sets). *On any compact $K = [-M, M]$, the vector field $f(\cdot; r_{t-1}^2)$ is Lipschitz continuous in $x$ with constant*

$$L_f(M) \leq \underbrace{L_\kappa\left(M + |\theta| + b\right) + \kappa_{\max}\left(L_\theta + 1\right)}_{\text{mean reversion term}} + \underbrace{L_{\text{jump}}\left(1 + \Lambda(M)\right)}_{\text{expected jump term}}.$$

*The exogenous clustering term $\gamma_+ r_{t-1}^2$ does not affect the Lipschitz constant in $x$.*

*Proof.* Write $f(x) = g(x)h(x) + c + q(x)$ where $g = \kappa_{\text{eff}}$, $h = \theta_{\text{eff}} - x$, $c = \gamma_+ r_{t-1}^2$ (constant in $x$), and $q = \lambda\, J$. For any $x_1, x_2 \in K$,

$$|g_1 h_1 - g_2 h_2| \leq |g_1 - g_2|\, |h_1| + |g_2|\, |h_1 - h_2|.$$

Using $|h(x)| \leq |\theta_{\text{eff}}(x)| + |x| \leq |\theta| + b + M$ and $|h_1 - h_2| \leq (L_\theta + 1)|x_1 - x_2|$, plus $|g_1 - g_2| \leq L_\kappa|x_1 - x_2|$ and $|g_2| \leq \kappa_{\max}$, we obtain the first bracket. For $q = \lambda J$,

$$|q_1 - q_2| \leq |\lambda_1 - \lambda_2|\, |J_1| + |\lambda_2|\, |J_1 - J_2| \leq \left(L_{\text{jump}} \cdot 1 + \Lambda(M)\, L_{\text{jump}}\right)|x_1 - x_2|.$$

Summing the bounds yields the stated $L_f(M)$. $\qquad\square$

**Theorem A.2** (Linear growth and global existence)**.** *There exist constants $C_0(M), C_1 \geq 0$ such that*

$$|f(x; r_{t-1}^2)| \leq C_0(M) + C_1 |x|, \qquad \forall x \in \mathbb{R}.$$

*One admissible choice is*

$$C_1 = \kappa_{\max} + L_{\mathrm{jump}}, \qquad C_0(M) = \kappa_{\max}(|\theta| + b) + |\gamma_+| \, |r_{t-1}^2| + \log 2 + |\ell(0)| + \varepsilon,$$

*using $\lambda(x) \leq \log 2 + |\ell(0)| + L_{\mathrm{jump}}|x| + \varepsilon$. Consequently, by Picard–Lindelöf and Grönwall, the IVP admits a unique solution globally in time.*

## A.2 Convergence of RK4 for Our Constrained Neural Mean–Reversion with Clustering and Jumps

We study our model

$$\dot{x}(t) = f\big(x(t), u(t)\big) = \underbrace{\kappa_{\mathrm{eff}}(x) \big(\theta_{\mathrm{eff}}(x) - x\big)}_{\text{mean reversion}} + \underbrace{\gamma_+ \, u(t)}_{\text{vol. clustering}} + \underbrace{\lambda(x) \, J(x)}_{\text{expected jumps}},$$

with

$$\kappa_{\mathrm{eff}}(x) = \kappa_0 \, \exp\big(s \, \tanh(k(x))\big), \qquad \theta_{\mathrm{eff}}(x) = \theta + b \, \tanh(\vartheta(x)),$$

$$\lambda(x) = \mathrm{softplus}\big(\ell(x)\big) + \varepsilon, \qquad J(x) = \tanh\big(j(x)\big),$$

where $k, \vartheta, \ell, j$ are scalar outputs of MLPs with exact GELU activations, and $u(t) = r_{t-1}^2$ is a bounded exogenous regressor. In our solver, we hold $u(t)$ piecewise–constant on each RK4 step (matching our experimental procedure).

Let $[0, T]$ be fixed and $K \subset \mathbb{R}$ be a compact set that contains the true trajectory and all RK4 stage values (for $h$ small enough).

**Hypotheses we verify.** - (H1) Uniform Lipschitz in $x$ on $K$: by our Lipschitz result in Appendix A.1, there exists $L \geq 0$ such that $|f(x_1, u) - f(x_2, u)| \leq L \, |x_1 - x_2|$ for all $x_1, x_2 \in K$ and admissible $u$. - (H2) Smoothness in $x$ on $K$: $\exp, \tanh, \mathrm{softplus}$ and exact GELU are $C^\infty$; compositions with affine maps preserve $C^\infty$, so $f(\cdot, u) \in C^\infty(K)$ uniformly in $u$. Hence $\partial_x^m f(\cdot, u)$ are bounded on $K$ for $m = 0, \dots, 4$. - (H3) Stage confinement: since $f$ is bounded on $K$, there exists $h_0 > 0$ such that for $0 < h \leq h_0$ all RK4 stages remain in $K$.

We apply classical RK4 with step $h > 0$, freezing $u(t) \equiv u_n$ on $[t_n, t_{n+1}]$, $t_n = nh$:

$$k_1 = f(x_n, u_n), \quad k_2 = f\big(x_n + \tfrac{h}{2}k_1, u_n\big), \quad k_3 = f\big(x_n + \tfrac{h}{2}k_2, u_n\big),$$

$$k_4 = f\big(x_n + hk_3, u_n\big), \qquad x_{n+1} = x_n + \tfrac{h}{6}(k_1 + 2k_2 + 2k_3 + k_4).$$

**Lemma A.3** (Local truncation error (order 5))**.** *There exists $C_{\mathrm{loc}} = C_{\mathrm{loc}}(K, f)$ such that, for all $x \in K$ and admissible (frozen) $u$,*

$$\big\| \varphi_h(x, u) - \Phi_h(x, u) \big\| \leq C_{\mathrm{loc}} \, h^5,$$

*where $\varphi_h$ is the exact time-h flow of $x' = f(x, u)$ and $\Phi_h$ is one RK4 step.*

*Proof.* By (H2), $f(\cdot, u) \in C^4(K)$ uniformly in $u$. The Taylor–Butcher expansion and RK4 order conditions give a uniform $\mathcal{O}(h^5)$ remainder. (H3) guarantees stage evaluations stay in $K$ where the bounds apply. $\square$

**Lemma A.4** (One–step Lipschitz stability)**.** *There exist $c_1 = c_1(K, f)$ and $h_1 > 0$ such that, for $0 < h \leq h_1$ and all $x, y \in K$ and admissible $u$,*

$$\big\| \Phi_h(x, u) - \Phi_h(y, u) \big\| \leq (1 + c_1 h) \, \|x - y\|.$$

*Proof.* Write $\Phi_h(x, u) = x + h \, \Psi_h(x, u)$ with stage–averaged slope $\Psi_h$. By (H1)–(H2), $\Psi_h(\cdot, u)$ is Lipschitz in $x$ with constant $L + \mathcal{O}(h)$ uniformly on $K$. Hence $\|\Phi_h(x, u) - \Phi_h(y, u)\| \leq (1 + h(L + C'))\|x - y\|$ for some $C'$, proving the claim. $\square$

**Theorem A.5** (Global $\mathcal{O}(h^4)$ error for our model). *Under* (H1)–(H3) *there exists* $C = C(K, f)$, *independent of* $h$ *and* $n$, *such that*

$$\max_{0 \leq n \leq N} \left\| x(t_n) - x_n \right\| \ \leq \ C \, e^{LT} \, h^4, \qquad N = \lfloor T/h \rfloor.$$

*Proof.* On $[t_n, t_{n+1}]$ with frozen $u_n$, the exact solution satisfies $x(t_{n+1}) = \varphi_h(x(t_n), u_n)$, so

$$e_{n+1} := x(t_{n+1}) - x_{n+1} = \left[ \Phi_h(x(t_n), u_n) - \Phi_h(x_n, u_n) \right] + \tau_{n+1}, \qquad \|\tau_{n+1}\| \leq C_{\mathrm{loc}} h^5.$$

By Lemma A.4, $\|e_{n+1}\| \leq (1 + c_1 h)\|e_n\| + C_{\mathrm{loc}} h^5$. Discrete Grönwall yields $\|e_n\| \leq C e^{c_1 t_n} h^4 \leq C e^{LT} h^4$, since $c_1 \geq L$ and $e_0 = 0$. $\qquad\square$

**Why the hypotheses hold for our architecture.** - Mean reversion term: $\kappa_{\mathrm{eff}}$ and $\theta_{\mathrm{eff}}$ are $C^\infty$ compositions of $\exp, \tanh$ and affine maps; on $K$ they and their first four derivatives are bounded. Our Lipschitz constant $L$ on $K$ was established in Appendix A.1. - Clustering term: $\gamma_+ \, u(t)$ is constant in $x$ and bounded in $t$, so it does not affect Lipschitz in $x$ nor the order of the method. - Jump term: $\lambda = \mathrm{softplus} \circ \ell$ and $J = \tanh \circ j$ are $C^\infty$; their products and derivatives are bounded on $K$. - Stage confinement: since $f$ is bounded on $K$, taking $h \leq h_0(K, f)$ ensures stage values remain in $K$.

### A.3 Datasets

We evaluate on the following 10 indices spanning major global markets and asset classes:

- **Developed Market Equities:** S&P 500 (`^GSPC`), NASDAQ (`^IXIC`), Dow Jones (`^DJI`)
- **Emerging Market Equities:** NIFTY50 (`^NSEI`)
- **Small/Large Cap:** Russell 1000 (`^RUI`), Russell 2000 (`^RUT`)
- **Government Bonds:** iShares Core U.S. Aggregate Bond ETF (`AGG`)
- **Cryptocurrency:** Bitcoin (`BTC-USD`), Ethereum (`ETH-USD`)
- **International:** FTSE 100 (`^FTSE`)

Daily OHLC prices are obtained via `yfinance`. Realized volatility is estimated using the Garman–Klass estimator (Eq. 1), and we model log-volatility $x_t = \ln(\sigma_t)$. Evaluation period: August 2005 to August 2025.

### A.4 Evaluation Protocol

**Rolling window configuration.** We use the following strictly temporal configuration:

- **Train:** 63 trading days ($\approx$3 months)
- **Validation:** 20 days
- **Test:** 30 days
- **Stride:** 30 days between consecutive window start dates

We selected the training window length via a small hyperparameter sweep over candidate lookback periods 7, 21, 63, 126, 252 trading days and evaluated normalized MSE on held-out rolling windows across all assets. The 63-day window produced the lowest aggregate NRMSE across assets. Intuitively, longer windows (126, 252) often span multiple volatility regimes, causing learned modulators to average over heterogeneous dynamics and thereby reduce predictive utility; visually, parameter trajectories estimated from 126/252-day fits show piecewise regime structure that a single parameter set cannot capture reliably. A short window (7, 21) lacks sufficient information to estimate even the compact neural modulators stably. Thus, the 63-day choice represents an empirical tradeoff between enough data to fit modulators and narrowness of the regime assumed.

**Walk-forward guarantee of no data leakage.** The walk-forward protocol ensures strictly out-of-sample testing by moving forward in time only. For each window, we:

1. Train a model on the training segment

2. Validate on the validation segment

3. Make predictions on the test segment

4. Roll forward by stride days and repeat

Critically, **we never revisit past test periods.** Once predictions are made for a test segment, that window is complete. Even though future training windows may include data that was previously used as test data, we never re-predict those periods with the benefit of future information the predictions have already been made and recorded using only past data. This is the defining property of walk-forward validation: we only predict forward in time, never backward.

For a train size of 63 days, validation size of 20 days, test size of 30 days, and stride of 30 days:

- Window 1: Train on days 1-63, Val on days 64-83, Test on days 84-113

- Window 2: Train on days 31-93, Val on days 94-113, Test on days 114-143

- Window 3: Train on days 61-123, Val on days 124-143, Test on days 144-173

By Window 3, days 84-113 (Window 1's test set) now appear in the training data. However, predictions for days 84-113 were already made in Window 1 using only data up to day 83. We do not re-predict them. This ensures that every prediction uses only information available at the time the prediction would have been made in real-world deployment.

**Forecast horizons and autoregressive rollout.** We evaluate at $h \in \{1, 5, 10, 20\}$ days ahead. For $h = 20$, each 30-day test window yields 10 independent predictions (days 1-20, 2-21, ..., 10-29 of the test period).

All predictions are made using **strictly autoregressive rollout with no future data leakage**. When predicting at time $t$ for horizon $h$, the model uses only information available up to time $t$. Critically:

- For multi-step predictions ($h > 1$), the model recursively feeds its own predictions forward

- No ground-truth values from $t + 1$ to $t + h - 1$ are provided during the rollout

- Exogenous variables (e.g., squared returns $r_{t-1}^2$) use only strictly causal values: when predicting at time $t$, we use $r_{t-1}^2$ (available), never $r_t^2$ or future returns

- At prediction time $t$ for target $t + h$, the model has no access to any data from time $t + 1$ onward

This ensures realistic evaluation: predictions are made exactly as they would be in real-world deployment, where future information is unavailable.

**Example of autoregressive rollout.** To predict log-volatility at $t + 20$ starting from time $t$:

1. Input: observed log-volatilities up to $t$, exogenous features up to $t - 1$ (e.g., $r_{t-1}^2$)

2. Predict $\hat{x}_{t+1}$ using the ODE integrated from $t$ to $t + 1$

3. Predict $\hat{x}_{t+2}$ using $\hat{x}_{t+1}$ as input (not ground truth $x_{t+1}$)

4. Continue rolling forward: $\hat{x}_{t+3}$ from $\hat{x}_{t+2}$, ..., $\hat{x}_{t+20}$ from $\hat{x}_{t+19}$

5. Exogenous features at each step use only past information (e.g., at step $t + k$, use predicted $\hat{r}^2_{t+k-1}$ derived from $\hat{x}_{t+k-1}$)

This protocol tests the model's ability to maintain stability under compounding error, the key challenge in multi-step forecasting.

### A.5 Training Configuration

Unless stated otherwise, we use:

- **Seeds:** 42, 43, 44 (results aggregated across all 3 seeds and 10 assets)

- **Horizon in loss:** $n_{\text{steps}} = 20$

- **Batch size:** 50

- **Networks:** Small MLP modulators with GELU activations
  - Hidden sizes tested: [64,64], [64,64,64], [64,64,64,64], [16,16],[32,32]
  - Selected: [64,64] (no further improvement with deeper architectures)

- **Optimizer:** Adam with learning rate $1 \times 10^{-3}$

- **Epochs:** 50 (with early stopping on validation loss)

- **ODE Integrator:** RK4 with fixed step size

- **Constraints:** Softplus for mean-reversion rate ($\kappa_+ > 0$), bounded shifts for long-run level ($\theta_{\text{eff}}$)

### A.6 Compute Environment

Experiments were run on Linux/macOS with Python 3.10–3.11. GPU was optional; CPU results were consistent modulo runtime.

**Core libraries:**

- PyTorch

- Neuromancer for ODE integration, systems, and losses

- NumPy $\geq 1.26$, SciPy $\geq 1.11$, Pandas $\geq 2.0$, Matplotlib $\geq 3.8$

### A.7 Evaluation Metrics

**NRMSE.** Normalized Root Mean Squared Error, normalized by the standard deviation of observed volatility (scale-independent metric, lower is better).

**Statistical significance.** We report Diebold–Mariano tests (Fisher combined across assets) to compare model pairs using two-sided squared-error differentials. No multiple-comparison correction is applied. Combined p-values are reported in Table 3.

### A.8 Econometric Baselines

GARCH-family models (GARCH, FIGARCH) were estimated using the `arch` Python library. Models were refit in each rolling window via maximum likelihood. Convergence was verified; the best-performing configuration after convergence is reported.

## A.9 Per-Asset Performance Metrics with Regime Breakdown

This section presents detailed per-asset performance metrics across all forecast horizons ($h \in \{1, 5, 10, 20\}$), broken down by volatility regime. For each asset-model combination, we report 12 metrics: MSE , MAE, QLIKE, and sMAPE for (1) aggregated data, (2) low volatility regime, and (3) high volatility regime. All values are rounded to 2 decimal places. **Bold values indicate the best-performing model** for each metric-regime combination within each asset.

**Regime definition:** Per asset, classify volatility using a rolling Z-score of realized volatility computed over a centered rolling window (63 steps). Low-vol if z < -0.5; high-vol if z > +0.5; observations in between are not used for regime slices.

Table 6: Short Horizon ($h = 1$) Performance: Crypto & Bonds

| Asset | Model | Aggregated | | | | Low Vol | | | | High Vol | | | |
|---|---|---|---|---|---|---|---|---|---|---|---|---|---|
| | | MSE | MAE | QL | sM | MSE | MAE | QL | sM | MSE | MAE | QL | sM |
| *Crypto* | | | | | | | | | | | | | |
| Bitcoin | RW | 0.37 | 0.48 | 1.07 | 12.55 | 0.43 | 0.53 | 0.58 | 12.70 | 0.48 | 0.57 | 2.49 | 16.15 |
| | ODE-MR | 0.35 | 0.48 | 1.35 | 12.35 | 0.44 | 0.55 | **0.27** | 13.42 | 0.84 | 0.86 | 4.08 | 23.68 |
| | Vanilla Neural ODE | 0.35 | 0.47 | 0.98 | 12.16 | **0.42** | 0.52 | 0.55 | 12.48 | 0.46 | 0.56 | 2.31 | 15.81 |
| | UDE-MR | 0.37 | 0.41 | 0.83 | 10.85 | 0.43 | 0.46 | 0.40 | **7.42** | 0.48 | 0.52 | 0.42 | 17.17 |
| | UDE-MRVC | **0.33** | 0.40 | 0.80 | 10.69 | **0.42** | 0.45 | 0.40 | **7.42** | 0.47 | 0.51 | 0.40 | 17.17 |
| | UDE-MRVCJ | 0.36 | **0.39** | **0.79** | **10.68** | 0.43 | 0.46 | 0.39 | 10.42 | 0.46 | **0.50** | 0.38 | 16.95 |
| | GARCH | 0.90 | 0.78 | 0.92 | 21.80 | 1.75 | 1.23 | 1.59 | 32.47 | 0.45 | 0.56 | 0.54 | 12.83 |
| | EWMA | 0.79 | 0.72 | 0.85 | 20.00 | 1.56 | 1.16 | 1.46 | 30.28 | 0.44 | 0.55 | 0.52 | 12.58 |
| | FIGARCH | 0.82 | 0.73 | 0.86 | 20.29 | 1.63 | 1.18 | 1.49 | 30.84 | **0.43** | 0.54 | 0.50 | **12.35** |
| | MertonJD | 0.84 | 0.76 | 0.87 | 20.70 | 1.73 | 1.24 | 1.60 | 32.39 | 0.84 | 0.94 | **0.24** | 18.21 |
| | LSTM | 0.55 | 0.59 | 1.92 | 15.10 | 0.71 | 0.70 | 0.79 | 17.12 | 0.73 | 0.67 | 5.14 | 18.39 |
| | GRU | 0.48 | 0.55 | 1.08 | 14.24 | 0.83 | 0.78 | 0.83 | 18.96 | 0.46 | 0.52 | 2.43 | 14.87 |
| | TimesFM | 0.35 | 0.46 | 1.00 | 11.88 | 0.45 | 0.53 | 0.52 | 12.79 | 0.52 | 0.60 | 2.65 | 16.85 |
| | TimesFM FS | 0.34 | 0.45 | 0.95 | 11.60 | 0.43 | 0.52 | 0.49 | 12.28 | 0.51 | 0.60 | 2.54 | 16.88 |
| Ethereum | RW | 0.31 | 0.44 | 0.86 | 12.49 | 0.34 | 0.48 | 0.50 | 12.32 | 0.45 | 0.55 | 1.96 | 16.85 |
| | ODE-MR | 0.44 | 0.55 | 1.94 | 14.89 | 0.37 | 0.52 | **0.27** | 13.54 | 1.06 | 0.98 | 5.60 | 28.37 |
| | Vanilla Neural ODE | 0.29 | 0.43 | 0.77 | 12.01 | 0.33 | 0.47 | 0.46 | 12.15 | 0.41 | 0.53 | 1.77 | 16.34 |
| | UDE-MR | 0.31 | 0.31 | 0.71 | 10.89 | 0.34 | **0.28** | 0.35 | **5.39** | 0.44 | 0.44 | 0.31 | 17.49 |
| | UDE-MRVC | 0.31 | 0.31 | 0.69 | 10.87 | 0.34 | 0.34 | 0.34 | 10.23 | 0.44 | 0.44 | 0.31 | 17.40 |
| | UDE-MRVCJ | 0.30 | **0.30** | 0.63 | **10.52** | 0.34 | **0.28** | 0.33 | 6.74 | 0.43 | **0.43** | 0.31 | 17.22 |
| | GARCH | 0.73 | 0.70 | 0.78 | 21.08 | 1.44 | 1.11 | 1.38 | 31.80 | 0.42 | 0.54 | 0.48 | 13.65 |
| | EWMA | 0.65 | 0.66 | 0.72 | 19.61 | 1.28 | 1.05 | 1.26 | 29.64 | 0.41 | 0.53 | 0.46 | 13.28 |
| | FIGARCH | 0.64 | 0.65 | 0.71 | 19.41 | 1.28 | 1.04 | 1.26 | 29.56 | 0.40 | 0.52 | 0.44 | 12.98 |
| | MertonJD | 0.69 | 0.69 | 0.75 | 20.30 | 1.41 | 1.12 | 1.38 | 31.52 | 0.72 | 0.86 | **0.25** | **9.32** |
| | LSTM | 0.40 | 0.50 | 0.99 | 13.90 | 0.58 | 0.63 | 0.63 | 16.52 | 0.49 | 0.58 | 2.38 | 17.50 |
| | GRU | 0.35 | 0.47 | 0.68 | 13.06 | 0.30 | 0.44 | 0.42 | 11.48 | **0.35** | 0.49 | 1.43 | 15.31 |
| | TimesFM | 0.27 | 0.41 | 0.69 | 11.40 | 0.36 | 0.49 | 0.43 | 12.49 | 0.42 | 0.56 | 1.77 | 16.98 |
| | TimesFM FS | **0.26** | 0.40 | **0.62** | 11.15 | 0.33 | 0.47 | 0.40 | 11.94 | 0.42 | 0.56 | 1.75 | 17.05 |
| *Bonds* | | | | | | | | | | | | | |
| AGG | RW | 0.24 | 0.39 | 0.46 | 6.17 | 0.55 | 0.41 | 0.24 | 6.29 | 0.35 | 0.48 | 1.09 | 7.88 |
| | ODE-MR | 0.26 | 0.42 | 0.44 | 6.55 | 0.38 | 0.50 | 0.32 | 7.70 | 0.31 | 0.46 | 0.58 | 7.50 |
| | Vanilla Neural ODE | 0.24 | 0.39 | 0.43 | 6.11 | 0.27 | 0.41 | 0.24 | 6.32 | 0.33 | 0.47 | 1.06 | 7.76 |
| | UDE-MR | **0.23** | **0.23** | **0.38** | **5.45** | **0.26** | **0.26** | **0.20** | 5.85 | 0.34 | **0.34** | 0.53 | 7.20 |
| | UDE-MRVC | 0.24 | 0.24 | 0.40 | 5.85 | **0.26** | **0.26** | 0.26 | 5.63 | 0.35 | 0.35 | **0.47** | **6.35** |
| | UDE-MRVCJ | 0.24 | 0.24 | 0.43 | 6.30 | **0.26** | **0.26** | 0.31 | 5.58 | 0.35 | 0.35 | 0.55 | 7.00 |
| | GARCH | 1.27 | 1.00 | 1.16 | 17.14 | 2.10 | 1.37 | 1.64 | 22.75 | 0.48 | 0.59 | 0.64 | 10.71 |
| | EWMA | 1.18 | 0.96 | 1.09 | 16.42 | 1.93 | 1.32 | 1.53 | 21.79 | 0.48 | 0.57 | 0.59 | 10.41 |
| | FIGARCH | 1.13 | 0.93 | 1.05 | 15.87 | 1.89 | 1.30 | 1.51 | 21.42 | 0.41 | 0.52 | 0.56 | 9.57 |
| | MertonJD | 1.35 | 1.05 | 1.22 | 17.91 | 2.23 | 1.44 | 1.75 | 23.87 | 0.51 | 0.61 | 0.61 | 11.13 |
| | LSTM | 0.51 | 0.57 | 0.58 | 9.13 | 0.89 | 0.82 | 0.73 | 12.82 | **0.23** | 0.39 | 0.64 | 6.55 |
| | GRU | 0.35 | 0.46 | 0.45 | 7.32 | 0.60 | 0.66 | 0.49 | 10.17 | 0.26 | 0.41 | 0.74 | 6.84 |
| | TimesFM | 0.27 | 0.40 | 0.43 | 6.36 | 0.38 | 0.50 | 0.33 | 7.61 | 0.25 | 0.40 | 0.76 | 6.73 |
| | TimesFM FS | 0.25 | 0.39 | 0.42 | 6.17 | 0.35 | 0.47 | 0.30 | 7.23 | 0.25 | 0.41 | 0.76 | 6.74 |

Table 7: Short Horizon ($h = 1$) Performance: Equities

| Asset | Model | Aggregated MSE | MAE | QL | sM | Low Vol MSE | MAE | QL | sM | High Vol MSE | MAE | QL | sM |
|---|---|---|---|---|---|---|---|---|---|---|---|---|---|
| *Equities* | | | | | | | | | | | | | |
| DowJones | RW | 0.22 | 0.37 | 0.54 | 7.36 | **0.23** | 0.39 | 0.34 | 7.22 | 0.28 | 0.44 | 1.05 | 9.06 |
| | ODE-MR | 0.29 | 0.44 | 0.39 | 8.63 | 0.58 | 0.70 | 0.69 | 13.37 | 0.29 | 0.46 | **0.25** | 9.49 |
| | Vanilla Neural ODE | **0.21** | 0.37 | 0.51 | 7.22 | **0.23** | 0.39 | **0.33** | 7.18 | 0.28 | 0.43 | 1.02 | 9.00 |
| | UDE-MR | 0.22 | 0.22 | **0.38** | 6.99 | **0.23** | 0.23 | 0.41 | 8.96 | 0.27 | **0.24** | 0.41 | **5.33** |
| | UDE-MRVC | **0.21** | **0.21** | **0.38** | 6.86 | **0.23** | 0.23 | 0.39 | **5.19** | 0.27 | 0.27 | 0.43 | 7.69 |
| | UDE-MRVCJ | 0.22 | 0.22 | 0.40 | **6.78** | **0.23** | 0.23 | 0.37 | 5.92 | 0.28 | 0.25 | 0.39 | **5.33** |
| | GARCH | 0.80 | 0.74 | 0.86 | 15.51 | 1.37 | 1.06 | 1.31 | 21.48 | 0.28 | 0.42 | 0.52 | 9.56 |
| | EWMA | 0.77 | 0.72 | 0.82 | 15.06 | 1.42 | 1.10 | 1.35 | 22.15 | 0.30 | 0.36 | 0.43 | 8.31 |
| | FIGARCH | 0.74 | 0.70 | 0.80 | 14.72 | 1.30 | 1.04 | 1.26 | 20.84 | 0.24 | 0.38 | 0.45 | 8.62 |
| | MertonJD | 0.85 | 0.77 | 0.89 | 16.26 | 1.56 | 1.16 | 1.47 | 23.59 | **0.23** | 0.39 | 0.37 | 8.58 |
| | LSTM | 0.47 | 0.54 | 1.96 | 10.51 | 0.40 | 0.50 | 0.58 | 9.49 | 0.63 | 0.68 | 3.21 | 13.97 |
| | GRU | 0.46 | 0.54 | 1.31 | 10.78 | 0.55 | 0.58 | 0.63 | 11.24 | 0.63 | 0.68 | 3.21 | 13.97 |
| | TimesFM | 0.27 | 0.40 | 0.75 | 7.94 | 0.34 | 0.48 | 0.44 | 9.02 | 0.36 | 0.48 | 1.70 | 9.92 |
| | TimesFM FS | 0.24 | 0.39 | 0.64 | 7.58 | 0.32 | 0.47 | 0.42 | 8.85 | 0.31 | 0.44 | 1.37 | 9.11 |
| FTSE | RW | 0.23 | 0.38 | 0.66 | 7.56 | 0.24 | 0.40 | 0.34 | 7.56 | 0.37 | 0.49 | 1.66 | 10.11 |
| | ODE-MR | 0.27 | 0.43 | **0.38** | 8.46 | 0.54 | 0.69 | 0.66 | 13.18 | 0.37 | 0.52 | 0.27 | 10.92 |
| | Vanilla Neural ODE | 0.23 | 0.38 | 0.63 | 7.45 | 0.24 | 0.40 | 0.34 | 7.60 | 0.35 | 0.48 | 1.59 | 9.92 |
| | UDE-MR | 0.23 | 0.30 | 0.41 | 6.83 | 0.24 | **0.24** | 0.38 | **5.35** | 0.36 | 0.36 | 0.36 | 8.82 |
| | UDE-MRVC | 0.23 | **0.23** | 0.40 | 6.57 | 0.24 | **0.24** | 0.33 | 6.14 | 0.36 | 0.36 | 0.37 | 8.64 |
| | UDE-MRVCJ | 0.23 | **0.23** | 0.39 | **6.44** | 0.24 | **0.24** | 0.32 | 7.69 | 0.37 | **0.26** | 0.37 | **5.10** |
| | GARCH | 0.58 | 0.62 | 0.66 | 12.97 | 1.03 | 0.92 | 1.05 | 18.49 | 0.19 | 0.35 | 0.35 | 8.01 |
| | EWMA | 0.52 | 0.58 | 0.61 | 12.10 | 0.93 | 0.88 | 0.98 | 17.55 | 0.18 | 0.33 | 0.36 | 7.40 |
| | FIGARCH | 0.52 | 0.59 | 0.61 | 12.27 | 0.95 | 0.89 | 1.00 | 17.86 | 0.16 | 0.33 | 0.33 | 7.34 |
| | MertonJD | 0.57 | 0.63 | 0.66 | 13.17 | 1.06 | 0.97 | 1.11 | 19.43 | 0.32 | 0.47 | **0.26** | 6.67 |
| | LSTM | 0.35 | 0.47 | 1.29 | 9.19 | **0.22** | 0.36 | 0.35 | 6.76 | 0.76 | 0.80 | 3.68 | 16.03 |
| | GRU | 0.35 | 0.47 | 1.03 | 9.25 | 0.32 | 0.43 | 0.42 | 8.17 | 0.70 | 0.61 | 2.75 | 14.08 |
| | TimesFM | 0.22 | 0.39 | 0.62 | 7.27 | 0.24 | 0.40 | 0.34 | 7.62 | **0.12** | 0.57 | 1.59 | 10.02 |
| | TimesFM FS | **0.20** | 0.36 | 0.54 | 7.00 | 0.24 | 0.41 | 0.33 | 7.64 | 0.34 | 0.47 | 1.36 | 9.74 |
| NASDAQ | RW | 0.25 | 0.38 | 0.61 | 7.83 | **0.23** | 0.39 | 0.34 | 7.68 | 0.32 | 0.46 | 1.31 | 10.16 |
| | ODE-MR | 0.29 | 0.38 | 0.52 | 8.80 | 0.43 | 0.59 | 0.68 | 11.66 | 0.35 | 0.51 | 0.85 | 10.52 |
| | Vanilla Neural ODE | 0.25 | 0.37 | 0.48 | 7.66 | 0.26 | 0.39 | 0.45 | 7.70 | 0.31 | 0.45 | 0.72 | 9.90 |
| | UDE-MR | 0.24 | 0.35 | 0.44 | 7.52 | 0.30 | 0.41 | 0.50 | 9.80 | 0.32 | 0.43 | 0.62 | 8.85 |
| | UDE-MRVC | **0.22** | **0.22** | **0.38** | 7.22 | **0.23** | 0.23 | **0.33** | **7.67** | 0.31 | **0.31** | 0.42 | **6.98** |
| | UDE-MRVCJ | 0.23 | 0.24 | **0.38** | **7.21** | **0.23** | 0.24 | 0.34 | 7.70 | 0.30 | 0.32 | 0.40 | 7.05 |
| | GARCH | 0.91 | 0.81 | 0.94 | 18.15 | 1.67 | 1.21 | 1.55 | 26.17 | 0.28 | 0.41 | 0.41 | 10.06 |
| | EWMA | 0.85 | 0.78 | 0.88 | 17.23 | 1.57 | 1.17 | 1.48 | 25.16 | 0.25 | 0.38 | 0.39 | 9.31 |
| | FIGARCH | 0.82 | 0.77 | 0.86 | 16.95 | 1.54 | 1.16 | 1.45 | 24.84 | **0.24** | 0.38 | **0.37** | 9.11 |
| | MertonJD | 0.91 | 0.82 | 0.94 | 18.16 | 1.70 | 1.24 | 1.59 | 26.56 | 0.25 | 0.41 | **0.37** | 9.91 |
| | LSTM | 0.52 | 0.56 | 2.40 | 11.51 | 0.43 | 0.53 | 0.70 | 10.43 | 0.90 | 0.80 | 5.91 | 17.10 |
| | GRU | 0.54 | 0.59 | 1.87 | 12.28 | 0.64 | 0.68 | 0.85 | 13.87 | 0.66 | 0.63 | 4.22 | 13.63 |
| | TimesFM | 0.29 | 0.42 | 0.91 | 8.61 | 0.37 | 0.52 | 0.51 | 10.18 | 0.38 | 0.48 | 1.97 | 10.56 |
| | TimesFM FS | 0.27 | 0.40 | 0.80 | 8.34 | 0.36 | 0.51 | 0.49 | 10.02 | 0.34 | 0.46 | 1.71 | 10.07 |
| NIFTY50 | RW | 0.22 | 0.37 | 0.57 | 7.30 | 0.25 | 0.39 | 0.34 | 7.40 | 0.33 | 0.46 | 1.29 | 9.53 |
| | ODE-MR | 0.25 | 0.41 | **0.35** | 8.10 | 0.52 | 0.67 | 0.63 | 12.79 | 0.32 | 0.51 | **0.22** | 10.34 |
| | Vanilla Neural ODE | **0.21** | 0.36 | 0.54 | 7.18 | **0.24** | 0.39 | 0.33 | 7.37 | 0.32 | 0.46 | 1.24 | 9.39 |
| | UDE-MR | 0.22 | 0.22 | 0.37 | 6.48 | 0.25 | **0.25** | 0.35 | 8.17 | 0.33 | **0.23** | 0.43 | **5.05** |
| | UDE-MRVC | 0.22 | 0.22 | 0.36 | 6.44 | 0.25 | **0.25** | 0.33 | **5.01** | 0.32 | 0.32 | 0.44 | 7.92 |
| | UDE-MRVCJ | **0.21** | **0.21** | 0.36 | **6.31** | 0.25 | **0.25** | 0.32 | 7.73 | 0.32 | 0.32 | 0.41 | 7.61 |
| | GARCH | 0.84 | 0.77 | 0.87 | 16.47 | 1.43 | 1.11 | 1.37 | 22.77 | 0.31 | 0.41 | 0.40 | 9.54 |
| | EWMA | 0.81 | 0.75 | 0.84 | 15.91 | 1.41 | 1.10 | 1.35 | 22.50 | 0.37 | 0.39 | 0.38 | 9.04 |
| | FIGARCH | 0.78 | 0.74 | 0.82 | 15.78 | 1.37 | 1.09 | 1.33 | 22.19 | 0.28 | 0.39 | 0.37 | 9.14 |
| | MertonJD | 0.86 | 0.80 | 0.91 | 17.05 | 1.49 | 1.15 | 1.44 | 23.62 | 0.28 | 0.43 | 0.40 | 9.81 |
| | LSTM | 0.40 | 0.50 | 1.37 | 9.87 | 0.28 | 0.37 | 0.40 | 7.14 | 0.75 | 0.80 | 3.41 | 16.26 |
| | GRU | 0.43 | 0.52 | 1.21 | 10.41 | 0.43 | 0.48 | 0.52 | 9.37 | 0.63 | 0.70 | 2.81 | 14.49 |
| | TimesFM | 0.24 | 0.38 | 0.64 | 7.46 | 0.27 | 0.41 | 0.36 | 7.88 | 0.35 | 0.48 | 1.42 | 10.00 |
| | TimesFM FS | 0.22 | 0.36 | 0.58 | 7.21 | 0.27 | 0.41 | 0.35 | 7.85 | **0.23** | 0.31 | 1.27 | 5.09 |
| Russell1000 | RW | 0.35 | 0.46 | 0.95 | 9.11 | 0.39 | 0.50 | 0.48 | 9.71 | 0.50 | 0.59 | 2.25 | 12.01 |
| | ODE-MR | 0.48 | 0.59 | 0.58 | 11.92 | 0.95 | 0.94 | 1.06 | 18.55 | 0.53 | 0.71 | **0.14** | 14.33 |
| | Vanilla Neural ODE | 0.64 | 0.54 | 3.31 | 10.01 | 0.52 | 0.51 | 0.81 | 9.26 | 1.04 | 0.75 | 9.70 | 14.36 |
| | UDE-MR | 0.36 | 0.37 | 0.52 | 8.12 | 0.40 | 0.39 | 0.52 | 10.71 | 0.49 | **0.35** | 1.05 | **5.12** |
| | UDE-MRVC | 0.33 | **0.34** | **0.51** | 7.62 | 0.38 | 0.38 | 0.49 | 9.94 | 0.46 | 0.37 | 1.18 | 6.28 |
| | UDE-MRVCJ | 0.34 | 0.35 | 0.54 | 7.85 | 0.40 | 0.40 | 0.47 | 10.42 | 0.47 | 0.39 | 1.09 | 6.90 |
| | GARCH | 1.91 | 1.21 | 1.60 | 27.14 | 2.89 | 1.60 | 2.26 | 34.49 | 0.80 | 0.69 | 0.77 | 16.75 |
| | EWMA | 1.36 | 1.03 | 1.28 | 22.40 | 2.20 | 1.42 | 1.92 | 29.77 | 0.48 | 0.54 | 0.54 | 12.74 |
| | FIGARCH | 1.97 | 1.21 | 1.62 | 27.51 | 2.93 | 1.60 | 2.27 | 34.75 | 0.85 | 0.69 | 0.79 | 17.02 |
| | MertonJD | 1.31 | 1.03 | 1.27 | 22.22 | 2.13 | 1.41 | 1.90 | 29.45 | **0.42** | 0.53 | 0.50 | 12.31 |
| | LSTM | 0.37 | 0.48 | 1.37 | 9.31 | **0.20** | 0.30 | 0.28 | 5.67 | 0.79 | 0.83 | 3.81 | 16.67 |
| | GRU | 0.37 | 0.47 | 1.07 | 9.23 | 0.31 | 0.39 | 0.37 | 7.48 | 0.65 | 0.74 | 2.81 | 15.11 |
| | TimesFM | 0.32 | 0.45 | 0.93 | 8.74 | 0.30 | 0.42 | 0.38 | 7.85 | 0.55 | 0.66 | 2.39 | 13.38 |
| | TimesFM FS | **0.31** | 0.44 | 0.64 | 8.65 | 0.30 | 0.43 | 0.38 | 8.06 | 0.52 | 0.63 | 2.24 | 12.90 |
| Russell2000 | RW | 0.38 | 0.50 | 1.29 | 10.43 | 0.42 | 0.54 | **0.53** | 11.04 | 0.59 | 0.64 | 3.29 | 13.94 |
| | ODE-MR | **0.31** | 0.45 | **0.49** | 9.60 | 0.61 | 0.73 | 0.72 | 14.68 | 0.50 | 0.62 | 0.60 | 13.41 |
| | Vanilla Neural ODE | 0.81 | 0.61 | 9.52 | 12.37 | 0.64 | 0.58 | 2.72 | 11.36 | 1.49 | 0.86 | 26.08 | 17.82 |
| | UDE-MR | 0.39 | 0.39 | 0.63 | 8.90 | 0.42 | 0.41 | 0.55 | 10.91 | 0.58 | 0.48 | 1.44 | 10.68 |
| | UDE-MRVC | 0.36 | **0.37** | 0.61 | **8.54** | 0.40 | **0.40** | 0.54 | 9.92 | 0.55 | 0.40 | 1.46 | 8.22 |
| | UDE-MRVCJ | 0.37 | 0.38 | 0.63 | 8.68 | 0.43 | 0.43 | **0.53** | 10.05 | 0.52 | 0.42 | 1.45 | 8.81 |
| | GARCH | 1.27 | 0.98 | 1.22 | 22.98 | 2.18 | 1.41 | 1.90 | 31.47 | 0.46 | 0.55 | 0.58 | 14.16 |
| | EWMA | 1.03 | 0.87 | 1.05 | 20.01 | 1.87 | 1.29 | 1.69 | 28.58 | 0.32 | 0.45 | 0.50 | 11.20 |
| | FIGARCH | 1.18 | 0.94 | 1.13 | 21.82 | 2.06 | 1.36 | 1.82 | 30.43 | 0.37 | 0.49 | 0.47 | 12.42 |
| | MertonJD | 0.99 | 0.87 | 1.02 | 19.86 | 1.81 | 1.30 | 1.69 | 28.43 | 0.41 | 0.52 | **0.42** | 10.84 |
| | LSTM | 0.46 | 0.52 | 2.25 | 10.88 | **0.31** | 0.44 | 0.56 | **8.70** | **0.19** | **0.35** | 5.99 | **8.16** |
| | GRU | 0.47 | 0.55 | 1.73 | 11.58 | 0.53 | 0.59 | 0.72 | 12.21 | 0.70 | 0.68 | 4.19 | 14.90 |
| | TimesFM | 0.39 | 0.49 | 1.44 | 10.26 | 0.42 | 0.54 | 0.59 | 10.88 | 0.64 | 0.66 | 3.61 | 14.58 |
| | TimesFM FS | 0.37 | 0.48 | 1.33 | 10.08 | 0.42 | 0.54 | 0.58 | 10.96 | 0.58 | 0.62 | 3.33 | 13.78 |
| SP500 | RW | 0.34 | 0.39 | 0.65 | 7.66 | **0.25** | 0.39 | 0.37 | 7.31 | 0.32 | 0.45 | 1.33 | 9.45 |
| | ODE-MR | 0.32 | 0.46 | 0.43 | 9.05 | 0.63 | 0.74 | 0.74 | 13.88 | 0.32 | 0.45 | **0.27** | 9.23 |
| | Vanilla Neural ODE | **0.24** | 0.39 | 0.62 | 7.58 | **0.25** | 0.40 | **0.36** | 7.36 | 0.33 | 0.45 | 1.27 | 9.30 |
| | UDE-MR | **0.24** | **0.24** | 0.44 | 7.48 | **0.25** | **0.25** | 0.47 | 9.81 | 0.31 | **0.25** | 0.42 | **5.59** |
| | UDE-MRVC | **0.24** | **0.24** | 0.43 | 7.28 | **0.25** | **0.25** | 0.44 | 9.26 | 0.30 | **0.25** | 0.40 | **5.59** |
| | UDE-MRVCJ | **0.24** | **0.24** | 0.42 | **7.21** | **0.25** | **0.25** | 0.42 | **5.54** | 0.32 | 0.32 | 0.42 | 7.41 |
| | GARCH | 0.99 | 0.85 | 1.01 | 17.97 | 1.79 | 1.26 | 1.63 | 25.53 | **0.29** | 0.44 | 0.44 | 10.07 |
| | EWMA | 0.97 | 0.83 | 0.97 | 17.31 | 1.75 | 1.23 | 1.59 | 24.95 | 0.32 | 0.39 | 0.43 | 8.98 |
| | FIGARCH | 0.92 | 0.82 | 0.94 | 17.09 | 1.69 | 1.22 | 1.56 | 24.55 | 0.36 | 0.41 | 0.40 | 9.52 |
| | MertonJD | 1.05 | 0.88 | 1.04 | 18.38 | 1.92 | 1.31 | 1.73 | 26.61 | 0.33 | 0.42 | 0.39 | 9.58 |
| | LSTM | 0.52 | 0.56 | 2.26 | 11.02 | 0.46 | 0.54 | 0.68 | 10.18 | 0.86 | 0.78 | 5.80 | 15.91 |
| | GRU | 0.52 | 0.57 | 1.47 | 11.40 | 0.64 | 0.64 | 0.72 | 12.45 | 0.64 | 0.65 | 3.49 | 13.50 |
| | TimesFM | 0.30 | 0.43 | 0.86 | 8.41 | 0.39 | 0.52 | 0.50 | 9.73 | 0.37 | 0.46 | 1.87 | 9.81 |
| | TimesFM FS | 0.27 | 0.41 | 0.72 | 8.04 | 0.37 | 0.51 | 0.48 | 9.52 | 0.32 | 0.44 | 1.51 | 9.24 |

Table 8: Medium Horizon ($h = 5$) Performance: Crypto & Bonds

| Asset | Model | Aggregated | | | | Low Vol | | | | High Vol | | | |
|---|---|---|---|---|---|---|---|---|---|---|---|---|---|
| | | MSE | MAE | QL | sM | MSE | MAE | QL | sM | MSE | MAE | QL | sM |
| *Crypto* | | | | | | | | | | | | | |
| Bitcoin | RW | 0.51 | 0.57 | 2.11 | 14.60 | 0.59 | 0.62 | 0.71 | 15.03 | 0.72 | 0.69 | 6.06 | 19.00 |
| | ODE-MR | 0.59 | 0.64 | 3.00 | 16.04 | 0.51 | 0.61 | **0.42** | 14.30 | 1.45 | 1.16 | 9.15 | 30.78 |
| | Vanilla Neural ODE | 0.42 | 0.51 | 1.29 | 13.20 | 0.54 | 0.60 | 0.64 | 14.45 | 0.60 | 0.64 | 3.30 | 17.68 |
| | UDE-MR | 0.50 | 0.53 | 1.22 | 12.31 | 0.58 | 0.61 | 0.49 | 8.40 | 0.69 | 0.72 | 0.42 | 20.63 |
| | UDE-MRVC | 0.49 | 0.52 | 1.21 | 12.07 | 0.58 | 0.60 | 0.46 | 7.98 | 0.68 | 0.71 | 0.40 | 20.48 |
| | UDE-MRVCJ | 0.47 | 0.50 | 1.14 | **11.73** | 0.57 | 0.59 | 0.44 | 7.98 | 0.65 | 0.68 | 0.38 | 20.38 |
| | GARCH | 1.11 | 0.89 | 1.07 | 25.07 | 2.12 | 1.38 | 1.85 | 36.88 | 0.23 | 0.37 | 0.36 | 11.90 |
| | EWMA | 0.82 | 0.74 | **0.87** | 20.45 | 1.64 | 1.19 | 1.51 | 31.24 | 0.72 | **0.34** | 0.44 | 10.73 |
| | FIGARCH | 1.03 | 0.85 | 1.01 | 23.75 | 1.99 | 1.33 | 1.76 | 35.24 | **0.21** | 0.35 | 0.35 | 11.26 |
| | MertonJD | 0.85 | 0.76 | **0.87** | 20.83 | 1.74 | 1.25 | 1.61 | 32.58 | 1.24 | 1.19 | **0.24** | **8.30** |
| | LSTM | 0.55 | 0.59 | 1.95 | 15.10 | 0.71 | 0.70 | 0.79 | 17.01 | 0.74 | 0.67 | 5.28 | 18.40 |
| | GRU | 0.48 | 0.55 | 1.09 | 14.21 | 0.80 | 0.77 | 0.83 | 18.68 | 0.46 | 0.53 | 2.49 | 14.99 |
| | TimesFM | 0.41 | 0.50 | 1.29 | 12.90 | 0.54 | 0.60 | 0.59 | 14.25 | 0.65 | 0.68 | 3.63 | 18.90 |
| | TimesFM FS | **0.39** | **0.49** | 1.26 | 12.62 | **0.49** | **0.58** | 0.55 | **5.06** | 0.65 | 0.69 | 3.59 | 19.26 |
| Ethereum | RW | 0.41 | 0.51 | 1.24 | 14.35 | 0.46 | 0.55 | 0.57 | 14.63 | 0.59 | 0.63 | 3.12 | 19.00 |
| | ODE-MR | 0.90 | 0.82 | 5.57 | 21.41 | 0.36 | 0.47 | 0.63 | 12.57 | 1.99 | 1.37 | 15.38 | 37.45 |
| | Vanilla Neural ODE | 0.32 | 0.45 | 0.85 | 12.56 | 0.42 | 0.54 | 0.51 | 13.96 | 0.46 | 0.57 | 2.13 | 17.16 |
| | UDE-MR | 0.40 | 0.40 | 0.92 | 12.10 | 0.56 | 0.56 | **0.31** | 21.60 | 0.46 | 0.45 | 0.39 | 11.41 |
| | UDE-MRVC | 0.39 | 0.39 | 0.90 | 11.83 | 0.56 | 0.56 | **0.31** | 21.21 | 0.45 | 0.43 | 0.33 | 10.25 |
| | UDE-MRVCJ | 0.38 | **0.38** | 0.88 | **11.75** | 0.54 | 0.54 | 0.32 | 20.54 | 0.41 | **0.39** | 0.32 | **9.84** |
| | GARCH | 0.83 | 0.77 | 0.88 | 23.26 | 0.19 | **0.34** | 0.35 | 11.74 | 1.61 | 1.19 | 1.51 | 34.34 |
| | EWMA | 0.68 | 0.67 | 0.75 | 20.13 | **0.18** | 0.63 | 0.38 | **11.22** | 1.34 | 1.07 | 1.31 | 30.44 |
| | FIGARCH | 0.79 | 0.74 | 0.83 | 22.31 | 0.20 | **0.34** | 0.36 | 11.73 | 1.51 | 1.15 | 1.44 | 32.87 |
| | MertonJD | 0.70 | 0.69 | 0.75 | 20.44 | 1.43 | 1.12 | 1.39 | 31.68 | 1.03 | 1.09 | **0.26** | 14.66 |
| | LSTM | 0.39 | 0.50 | 0.97 | 13.83 | 0.59 | 0.63 | 0.63 | 16.66 | 0.48 | 0.58 | 2.30 | 17.33 |
| | GRU | 0.35 | 0.47 | 0.67 | 13.00 | 0.62 | 0.67 | 0.66 | 17.47 | **0.35** | 0.49 | 1.39 | 15.19 |
| | TimesFM | 0.32 | 0.45 | 0.86 | 12.42 | 0.39 | 0.50 | 0.45 | 12.88 | 0.51 | 0.62 | 2.28 | 18.83 |
| | TimesFM FS | **0.31** | 0.44 | **0.61** | 12.28 | 0.37 | 0.48 | 0.42 | 12.21 | 0.53 | 0.64 | 2.34 | 19.19 |
| *Bonds* | | | | | | | | | | | | | |
| AGG | RW | 0.29 | 0.42 | 0.57 | 6.63 | 0.32 | 0.45 | 0.29 | 6.88 | 0.46 | 0.55 | 1.51 | 8.97 |
| | ODE-MR | 0.32 | 0.45 | 0.58 | 6.80 | 0.36 | 0.48 | 0.32 | 7.35 | 0.44 | 0.54 | 1.42 | 8.70 |
| | Vanilla Neural ODE | 0.29 | 0.42 | 0.56 | 6.63 | 0.35 | 0.48 | 0.32 | 7.31 | 0.42 | 0.52 | 1.41 | 8.54 |
| | UDE-MR | 0.29 | 0.29 | 0.47 | 6.50 | **0.30** | **0.30** | **0.23** | **6.45** | 0.44 | 0.44 | 0.58 | 7.85 |
| | UDE-MRVC | 0.28 | **0.28** | 0.43 | **5.85** | 0.31 | 0.31 | 0.30 | 7.50 | 0.44 | 0.35 | **0.48** | 6.20 |
| | UDE-MRVCJ | 0.29 | 0.29 | 0.54 | 7.40 | 0.32 | 0.32 | 0.38 | 8.20 | 0.46 | 0.44 | 0.68 | 7.95 |
| | GARCH | 1.64 | 1.17 | 1.43 | 20.36 | 2.62 | 1.56 | 1.98 | 26.32 | 0.68 | 0.72 | 0.77 | 13.42 |
| | EWMA | 1.23 | 0.98 | 1.13 | 16.80 | 1.98 | 1.34 | 1.57 | 22.16 | 0.49 | 0.58 | 0.62 | 10.56 |
| | FIGARCH | 1.51 | 1.11 | 1.33 | 19.30 | 2.39 | 1.49 | 1.84 | 24.95 | 0.61 | 0.67 | 0.70 | 12.49 |
| | MertonJD | 1.36 | 1.05 | 1.23 | 18.00 | 2.25 | 1.45 | 1.76 | 24.01 | 0.52 | 0.62 | 0.62 | 11.19 |
| | LSTM | 0.50 | 0.57 | 0.59 | 9.12 | 0.89 | 0.82 | 0.74 | 12.84 | 0.24 | 0.40 | 0.67 | 6.69 |
| | GRU | 0.35 | 0.46 | 0.45 | 7.29 | 0.60 | 0.67 | 0.50 | 10.28 | 0.26 | 0.41 | 0.75 | 6.93 |
| | TimesFM | 0.28 | 0.41 | 0.45 | 6.54 | 0.44 | 0.55 | 0.37 | 8.38 | 0.18 | 0.33 | 0.50 | 5.59 |
| | TimesFM FS | **0.27** | 0.40 | 0.45 | 6.34 | 0.41 | 0.52 | 0.34 | 8.00 | **0.17** | **0.32** | **0.48** | **5.44** |

Table 9: Medium Horizon ($h = 5$) Performance: Equities

| Asset | Model | Aggregated | | | | Low Vol | | | | High Vol | | | |
|---|---|---|---|---|---|---|---|---|---|---|---|---|---|
| | | MSE | MAE | QL | sM | MSE | MAE | QL | sM | MSE | MAE | QL | sM |
| *Equities* | | | | | | | | | | | | | |
| DowJones | RW | 0.30 | 0.44 | 0.84 | 8.59 | 0.32 | 0.46 | 0.42 | 8.68 | 0.42 | 0.53 | 1.97 | 10.90 |
| | ODE-MR | 0.54 | 0.62 | 0.64 | 12.45 | 1.10 | 0.99 | 1.16 | 19.33 | 0.38 | 0.53 | **0.22** | 10.83 |
| | Vanilla Neural ODE | **0.27** | 0.42 | 0.73 | 8.25 | 0.31 | 0.46 | 0.41 | 8.69 | 0.40 | 0.52 | 1.73 | 10.76 |
| | UDE-MR | 0.29 | **0.29** | 0.58 | 8.68 | **0.31** | 0.31 | 0.62 | **5.06** | 0.41 | 0.41 | 0.51 | 9.49 |
| | UDE-MRVC | 0.29 | **0.29** | 0.53 | **6.59** | 0.32 | 0.32 | 0.57 | 11.35 | 0.39 | 0.37 | 0.49 | 7.86 |
| | UDE-MRVCJ | 0.30 | 0.30 | **0.52** | 8.36 | 0.33 | 0.33 | 0.56 | 11.03 | 0.41 | **0.29** | 0.45 | **5.87** |
| | GARCH | 0.92 | 0.81 | 0.97 | 17.18 | 1.61 | 1.18 | 1.50 | 24.05 | 0.32 | 0.44 | 0.56 | 10.24 |
| | EWMA | 0.80 | 0.73 | 0.84 | 15.43 | 1.47 | 1.12 | 1.39 | 22.64 | 0.25 | 0.38 | 0.43 | 8.65 |
| | FIGARCH | 0.83 | 0.76 | 0.89 | 16.04 | 1.52 | 1.15 | 1.43 | 23.16 | 0.26 | 0.39 | 0.46 | 9.07 |
| | MertonJD | 0.86 | 0.78 | 0.90 | 16.38 | 1.58 | 1.17 | 1.48 | 23.73 | **0.24** | 0.42 | 0.38 | 8.84 |
| | LSTM | 0.50 | 0.56 | 2.02 | 10.82 | 0.42 | 0.52 | 0.59 | 9.71 | 0.72 | 0.73 | 3.95 | 14.85 |
| | GRU | 0.45 | 0.54 | 1.27 | 10.74 | 0.53 | 0.57 | 0.61 | 11.04 | 0.63 | 0.69 | 3.09 | 14.08 |
| | TimesFM | 0.35 | 0.47 | 0.95 | 9.26 | 0.46 | 0.56 | 0.57 | 10.70 | 0.45 | 0.55 | 2.14 | 11.37 |
| | TimesFM FS | 0.34 | 0.46 | 0.88 | 9.11 | 0.46 | 0.57 | 0.56 | 10.75 | 0.41 | 0.52 | 1.93 | 10.82 |
| FTSE | RW | 0.27 | 0.41 | 0.71 | 8.10 | 0.31 | 0.44 | 0.42 | 8.45 | 0.39 | 0.52 | 1.69 | 10.66 |
| | ODE-MR | 0.49 | 0.60 | 0.59 | 12.01 | 0.97 | 0.94 | 1.06 | 18.40 | 0.36 | 0.53 | **0.18** | 11.19 |
| | Vanilla Neural ODE | 0.26 | 0.40 | 0.66 | 7.97 | 0.33 | 0.46 | 0.42 | 8.88 | 0.37 | 0.49 | 1.55 | 10.09 |
| | UDE-MR | 0.27 | 0.27 | 0.44 | 7.66 | 0.30 | **0.30** | 0.51 | 10.62 | 0.37 | **0.26** | 0.44 | **5.35** |
| | UDE-MRVC | 0.26 | **0.26** | 0.43 | **5.04** | 0.31 | 0.31 | 0.41 | 9.22 | 0.37 | **0.26** | 0.43 | **5.35** |
| | UDE-MRVCJ | 0.27 | 0.27 | **0.41** | 7.06 | 0.31 | 0.31 | 0.41 | 9.06 | 0.39 | 0.27 | 0.44 | 6.53 |
| | GARCH | 0.63 | 0.65 | 0.71 | 13.73 | 1.12 | 0.97 | 1.14 | 19.72 | 0.21 | 0.36 | 0.38 | 8.17 |
| | EWMA | 0.55 | 0.60 | 0.65 | 12.60 | 0.97 | 0.91 | 1.02 | 18.18 | 0.21 | 0.35 | 0.45 | 7.95 |
| | FIGARCH | 0.58 | 0.63 | 0.66 | 13.17 | 1.08 | 0.97 | 1.12 | 19.56 | **0.16** | 0.32 | 0.31 | 7.11 |
| | MertonJD | 0.58 | 0.64 | 0.68 | 13.36 | 1.08 | 0.98 | 1.14 | 19.75 | 0.29 | 0.47 | 0.31 | 7.06 |
| | LSTM | 0.34 | 0.47 | 1.28 | 9.18 | **0.23** | 0.36 | **0.35** | **6.82** | 0.75 | 0.78 | 3.62 | 15.71 |
| | GRU | 0.35 | 0.47 | 1.03 | 9.25 | 0.32 | 0.43 | 0.42 | 8.30 | 0.60 | 0.68 | 2.72 | 13.81 |
| | TimesFM | 0.26 | 0.40 | 0.71 | 7.92 | 0.30 | 0.45 | 0.40 | 8.46 | 0.41 | 0.54 | 1.75 | 11.10 |
| | TimesFM FS | **0.25** | 0.40 | 0.59 | 7.79 | 0.30 | 0.45 | 0.40 | 8.55 | 0.38 | 0.51 | 1.61 | 10.58 |
| NASDAQ | RW | 0.30 | 0.44 | 0.78 | 9.05 | 0.35 | 0.48 | 0.46 | 9.49 | 0.38 | 0.51 | 1.61 | 11.21 |
| | ODE-MR | 0.35 | 0.48 | **0.48** | 10.02 | 0.73 | 0.78 | 0.82 | 15.58 | 0.36 | 0.51 | 0.40 | 11.18 |
| | Vanilla Neural ODE | **0.28** | 0.42 | 0.70 | 8.67 | 0.35 | 0.48 | **0.44** | 9.52 | 0.34 | 0.49 | 1.41 | 10.69 |
| | UDE-MR | 0.29 | **0.29** | 0.59 | 8.96 | **0.34** | **0.34** | 0.61 | 12.55 | 0.36 | 0.35 | 0.49 | 8.32 |
| | UDE-MRVC | 0.29 | **0.29** | 0.54 | **5.49** | 0.35 | 0.35 | 0.57 | **6.57** | 0.36 | 0.36 | 0.49 | 9.94 |
| | UDE-MRVCJ | 0.30 | 0.30 | 0.53 | 8.77 | 0.35 | 0.35 | 0.57 | 11.79 | 0.37 | **0.31** | 0.45 | **7.22** |
| | GARCH | 1.11 | 0.91 | 1.10 | 20.66 | 2.02 | 1.36 | 1.80 | 29.60 | 0.36 | 0.48 | 0.49 | 11.75 |
| | EWMA | 0.88 | 0.79 | 0.91 | 17.64 | 1.64 | 1.21 | 1.53 | 25.87 | **0.28** | 0.40 | 0.42 | 9.70 |
| | FIGARCH | 1.05 | 0.88 | 1.05 | 19.80 | 1.93 | 1.32 | 1.74 | 28.62 | 0.35 | 0.46 | 0.48 | 11.27 |
| | MertonJD | 0.92 | 0.83 | 0.95 | 18.31 | 1.74 | 1.26 | 1.62 | 26.90 | 0.30 | 0.48 | **0.39** | 10.20 |
| | LSTM | 0.52 | 0.57 | 2.39 | 11.55 | 0.44 | 0.53 | 0.69 | 10.53 | 0.91 | 0.81 | 6.02 | 17.14 |
| | GRU | 0.54 | 0.60 | 1.86 | 12.33 | 0.65 | 0.69 | 0.84 | 14.05 | 0.69 | 0.66 | 4.29 | 14.09 |
| | TimesFM | 0.38 | 0.49 | 1.17 | 10.04 | 0.50 | 0.60 | 0.64 | 11.99 | 0.47 | 0.54 | 2.61 | 11.86 |
| | TimesFM FS | 0.36 | 0.48 | 1.06 | 9.81 | 0.49 | 0.60 | 0.63 | 11.96 | 0.33 | 0.48 | 2.32 | 10.75 |
| NIFTY50 | RW | 0.27 | 0.40 | 0.73 | 7.98 | 0.31 | 0.44 | 0.39 | 8.40 | 0.37 | 0.49 | 1.63 | 9.99 |
| | ODE-MR | 0.45 | 0.56 | 0.54 | 11.32 | 0.91 | 0.91 | 1.01 | 17.90 | 0.30 | 0.48 | **0.15** | 9.86 |
| | Vanilla Neural ODE | 0.30 | 0.39 | 0.58 | 7.67 | 0.30 | 0.44 | **0.38** | 8.49 | 0.35 | 0.47 | 1.61 | 9.46 |
| | UDE-MR | 0.27 | 0.27 | 0.41 | 7.25 | 0.30 | **0.30** | 0.47 | **6.64** | 0.36 | 0.25 | 0.43 | 5.13 |
| | UDE-MRVC | 0.26 | 0.26 | 0.39 | 7.13 | 0.31 | 0.31 | 0.43 | 9.55 | 0.35 | 0.25 | 0.42 | 5.05 |
| | UDE-MRVCJ | **0.25** | **0.25** | **0.38** | **7.08** | 0.31 | 0.31 | 0.42 | 9.34 | 0.34 | **0.24** | 0.33 | **5.01** |
| | GARCH | 1.05 | 0.89 | 1.04 | 19.19 | 1.81 | 1.26 | 1.63 | 26.16 | 0.34 | 0.47 | 0.44 | 11.00 |
| | EWMA | 0.83 | 0.76 | 0.85 | 16.23 | 1.43 | 1.11 | 1.36 | 22.70 | **0.28** | 0.40 | 0.37 | 9.38 |
| | FIGARCH | 0.97 | 0.86 | 0.99 | 18.36 | 1.70 | 1.22 | 1.56 | 25.34 | 0.30 | 0.45 | 0.41 | 10.40 |
| | MertonJD | 0.86 | 0.80 | 0.91 | 17.06 | 1.50 | 1.15 | 1.44 | 23.55 | **0.28** | 0.43 | 0.39 | 9.81 |
| | LSTM | 0.40 | 0.50 | 1.35 | 9.84 | **0.28** | 0.37 | 0.39 | 7.05 | 0.73 | 0.79 | 3.35 | 15.97 |
| | GRU | 0.43 | 0.52 | 1.19 | 10.36 | 0.43 | 0.47 | 0.51 | 9.29 | 0.63 | 0.70 | 2.75 | 14.33 |
| | TimesFM | 0.28 | 0.42 | 0.73 | 8.29 | 0.34 | 0.46 | 0.43 | 8.87 | 0.40 | 0.52 | 1.60 | 10.79 |
| | TimesFM FS | 0.27 | 0.40 | 0.65 | 8.02 | 0.33 | 0.47 | 0.43 | 8.97 | 0.34 | 0.49 | 1.38 | 10.01 |
| Russell1000 | RW | 0.28 | 0.41 | 0.74 | 8.07 | 0.30 | 0.43 | 0.39 | 8.14 | 0.43 | 0.55 | 1.80 | 11.24 |
| | ODE-MR | 0.48 | 0.59 | 0.58 | 11.92 | 0.95 | 0.94 | 1.05 | 18.38 | 0.48 | 0.65 | **0.17** | 13.35 |
| | Vanilla Neural ODE | **0.26** | 0.39 | 0.80 | **7.51** | 0.26 | 0.40 | 0.37 | 7.47 | 0.43 | 0.54 | 1.92 | 11.02 |
| | UDE-MR | 0.28 | 0.28 | 0.54 | 7.86 | 0.30 | 0.30 | 0.49 | 10.37 | 0.42 | 0.29 | 0.51 | 5.06 |
| | UDE-MRVC | 0.27 | **0.27** | 0.52 | 7.81 | 0.29 | **0.29** | 0.45 | 9.81 | 0.40 | **0.28** | 0.49 | 5.02 |
| | UDE-MRVCJ | 0.27 | **0.27** | 0.50 | 7.81 | 0.30 | 0.30 | 0.44 | 9.61 | 0.41 | **0.28** | 0.47 | **5.01** |
| | GARCH | 1.41 | 1.06 | 1.32 | 23.02 | 2.30 | 1.46 | 1.98 | 30.64 | 0.47 | 0.54 | 0.53 | 12.70 |
| | EWMA | 1.23 | 0.98 | 1.18 | 21.11 | 2.06 | 1.37 | 1.83 | 28.73 | 0.39 | 0.48 | 0.46 | 11.35 |
| | FIGARCH | 1.39 | 1.05 | 1.30 | 22.82 | 2.22 | 1.43 | 1.94 | 30.09 | 0.46 | 0.53 | 0.52 | 12.51 |
| | MertonJD | 1.29 | 1.02 | 1.25 | 22.00 | 2.15 | 1.42 | 1.92 | 29.70 | **0.38** | 0.51 | 0.47 | 11.66 |
| | LSTM | 0.38 | 0.49 | 1.46 | 9.48 | **0.22** | 0.32 | **0.35** | **5.98** | 0.84 | 0.85 | 4.19 | 17.23 |
| | GRU | 0.38 | 0.48 | 1.13 | 9.41 | 0.33 | 0.41 | 0.42 | 7.76 | 0.69 | 0.76 | 3.06 | 15.56 |
| | TimesFM | 0.30 | 0.43 | 0.82 | 8.46 | 0.32 | 0.45 | 0.42 | 8.52 | 0.49 | 0.59 | 2.10 | 12.31 |
| | TimesFM FS | 0.29 | 0.42 | 0.77 | 8.36 | 0.33 | 0.46 | 0.42 | 8.70 | 0.46 | 0.58 | 1.93 | 11.92 |
| Russell2000 | RW | 0.31 | 0.44 | 0.79 | 9.34 | 0.37 | 0.49 | 0.46 | 9.95 | 0.40 | 0.52 | 1.65 | 11.54 |
| | ODE-MR | **0.29** | 0.44 | **0.44** | 9.21 | 0.58 | 0.70 | 0.69 | 14.12 | 0.38 | 0.54 | 0.50 | 12.16 |
| | Vanilla Neural ODE | **0.29** | 0.43 | 0.62 | 8.99 | 0.32 | 0.46 | **0.44** | 9.24 | 0.41 | 0.54 | 1.69 | 11.91 |
| | UDE-MR | 0.31 | 0.31 | 0.54 | 8.57 | 0.36 | **0.36** | 0.51 | **5.14** | 0.39 | 0.39 | 0.47 | 10.04 |
| | UDE-MRVC | 0.30 | **0.30** | 0.53 | 8.53 | 0.36 | **0.36** | 0.51 | 11.36 | 0.38 | 0.37 | 0.45 | 8.54 |
| | UDE-MRVCJ | 0.30 | **0.30** | 0.50 | **8.33** | 0.37 | 0.37 | 0.50 | 11.14 | 0.37 | **0.32** | 0.47 | **7.44** |
| | GARCH | 1.07 | 0.90 | 1.08 | 20.79 | 1.87 | 1.31 | 1.71 | 28.83 | 0.33 | 0.46 | 0.48 | 11.69 |
| | EWMA | 0.90 | 0.81 | 0.94 | 18.50 | 1.63 | 1.21 | 1.54 | 26.42 | **0.25** | 0.39 | 0.39 | 9.66 |
| | FIGARCH | 0.95 | 0.85 | 0.99 | 19.44 | 1.71 | 1.25 | 1.61 | 27.46 | 0.27 | 0.41 | 0.40 | 10.28 |
| | MertonJD | 0.95 | 0.85 | 0.99 | 19.38 | 1.75 | 1.27 | 1.64 | 27.87 | 0.30 | 0.46 | **0.36** | 9.96 |
| | LSTM | 0.44 | 0.52 | 2.11 | 10.67 | **0.31** | 0.43 | 0.54 | 8.53 | 0.88 | 0.82 | 5.46 | 17.62 |
| | GRU | 0.46 | 0.53 | 1.60 | 11.27 | 0.52 | 0.59 | 0.70 | 12.11 | 0.64 | 0.65 | 3.74 | 14.18 |
| | TimesFM | 0.33 | 0.45 | 1.04 | 9.39 | 0.41 | 0.54 | 0.55 | 10.81 | 0.48 | 0.56 | 2.45 | 12.44 |
| | TimesFM FS | 0.31 | 0.44 | 0.95 | 9.22 | 0.40 | 0.54 | 0.54 | 10.87 | 0.42 | 0.33 | 2.22 | 11.74 |
| SP500 | RW | 0.35 | 0.47 | 0.98 | 9.21 | 0.38 | 0.50 | 0.49 | 9.30 | 0.45 | 0.55 | 2.08 | 11.26 |
| | ODE-MR | 0.62 | 0.66 | 0.70 | 13.07 | 1.24 | 1.05 | 1.26 | 20.35 | 0.36 | 0.48 | **0.26** | 10.24 |
| | Vanilla Neural ODE | 0.38 | 0.49 | 1.01 | 9.59 | 0.44 | 0.54 | 0.57 | 10.24 | 0.43 | 0.54 | 2.09 | 11.04 |
| | UDE-MR | 0.35 | 0.35 | 0.69 | 9.61 | **0.37** | **0.37** | 0.75 | 13.76 | 0.43 | **0.30** | 0.54 | **6.54** |
| | UDE-MRVC | **0.33** | **0.33** | 0.62 | 9.30 | **0.37** | **0.37** | 0.67 | **5.23** | 0.42 | 0.42 | 0.50 | 9.19 |
| | UDE-MRVCJ | 0.34 | 0.34 | **0.61** | **9.16** | 0.38 | 0.38 | 0.66 | 12.39 | 0.43 | 0.39 | 0.46 | 8.39 |
| | GARCH | 1.18 | 0.94 | 1.14 | 19.93 | 2.13 | 1.39 | 1.86 | 28.34 | 0.34 | 0.47 | 0.48 | 10.88 |
| | EWMA | 1.01 | 0.85 | 1.01 | 17.78 | 1.85 | 1.28 | 1.66 | 25.81 | **0.28** | 0.40 | 0.44 | 9.22 |
| | FIGARCH | 1.15 | 0.92 | 1.11 | 19.51 | 2.08 | 1.37 | 1.82 | 27.77 | 0.34 | 0.46 | 0.45 | 10.74 |
| | MertonJD | 1.06 | 0.89 | 1.05 | 18.56 | 1.96 | 1.33 | 1.76 | 26.91 | 0.29 | 0.44 | 0.40 | 9.68 |
| | LSTM | 0.52 | 0.56 | 2.23 | 11.01 | 0.46 | 0.54 | 0.67 | 10.20 | 0.88 | 0.80 | 5.72 | 16.28 |
| | GRU | 0.52 | 0.57 | 1.46 | 11.41 | 0.65 | 0.65 | 0.72 | 12.45 | 0.64 | 0.66 | 3.50 | 13.58 |
| | TimesFM | 0.41 | 0.51 | 1.10 | 9.98 | 0.56 | 0.63 | 0.67 | 11.96 | 0.46 | 0.55 | 2.40 | 11.30 |
| | TimesFM FS | 0.40 | 0.50 | 1.04 | 9.88 | 0.56 | 0.64 | 0.67 | 12.02 | 0.43 | 0.52 | 2.21 | 10.99 |

Table 10: Medium Horizon ($h = 10$) Performance: Crypto & Bonds

| Asset | Model | Aggregated | | | | Low Vol | | | | High Vol | | | |
|---|---|---|---|---|---|---|---|---|---|---|---|---|---|
| | | MSE | MAE | QL | sM | MSE | MAE | QL | sM | MSE | MAE | QL | sM |
| *Crypto* | | | | | | | | | | | | | |
| Bitcoin | RW | 0.57 | 0.60 | 2.28 | 15.38 | 0.69 | 0.68 | 0.78 | 16.72 | 1.24 | 1.17 | 0.54 | 14.53 |
| | ODE-MR | 0.60 | 0.64 | 3.04 | 16.10 | **0.50** | 0.62 | **0.44** | 14.46 | 0.92 | 0.87 | 9.12 | 18.43 |
| | Vanilla Neural ODE | 0.46 | 0.53 | 1.68 | 13.50 | 0.60 | 0.64 | 0.70 | 15.53 | 0.79 | 0.73 | 4.90 | 18.06 |
| | UDE-MR | 0.55 | 0.48 | 1.25 | 12.36 | 0.68 | 0.52 | 0.50 | 9.10 | 0.75 | 0.73 | 0.40 | 10.39 |
| | UDE-MRVC | 0.55 | 0.48 | 1.23 | 12.26 | 0.67 | 0.47 | 0.47 | **8.12** | 0.74 | 0.72 | 0.40 | 10.15 |
| | UDE-MRVCJ | 0.52 | **0.47** | 1.16 | **11.91** | 0.66 | **0.46** | 0.45 | **8.12** | **0.73** | **0.71** | 0.39 | **9.93** |
| | GARCH | 1.26 | 0.95 | 1.18 | 27.22 | 2.38 | 1.46 | 2.00 | 39.49 | 0.77 | 0.74 | 0.40 | 20.57 |
| | EWMA | 0.87 | 0.76 | 0.90 | 21.03 | 1.72 | 1.22 | 1.57 | 32.14 | 0.82 | 0.73 | 0.41 | 10.62 |
| | FIGARCH | 1.21 | 0.91 | 1.12 | 25.93 | 2.30 | 1.42 | 1.94 | 38.32 | 0.87 | 0.76 | **0.37** | 12.41 |
| | MertonJD | 0.86 | 0.77 | 0.88 | 21.02 | 1.77 | 1.26 | 1.63 | 32.86 | 0.94 | 0.82 | 0.41 | 13.45 |
| | LSTM | 0.54 | 0.59 | 1.94 | 15.00 | 0.70 | 0.70 | 0.79 | 16.72 | 1.26 | 0.95 | 5.15 | 18.27 |
| | GRU | 0.48 | 0.55 | 1.08 | 14.14 | 0.82 | 0.78 | 0.84 | 18.89 | 0.99 | 0.85 | 2.42 | 14.74 |
| | TimesFM | 0.43 | 0.52 | 1.34 | 13.24 | 0.58 | 0.63 | 0.63 | 14.99 | 0.89 | 0.78 | 3.61 | 18.43 |
| | TimesFM FS | **0.41** | 0.51 | **0.68** | 12.99 | 0.52 | 0.59 | 0.59 | 14.04 | 0.85 | 0.77 | 3.58 | 18.63 |
| Ethereum | RW | 0.46 | 0.54 | 1.44 | 15.14 | 0.55 | 0.60 | 0.64 | 16.00 | 1.01 | 1.07 | **0.25** | **9.73** |
| | ODE-MR | 0.91 | 0.83 | 5.62 | 21.48 | **0.37** | 0.48 | 0.66 | 12.61 | 0.57 | 0.57 | 15.23 | 21.28 |
| | Vanilla Neural ODE | 0.34 | 0.46 | 0.83 | 12.83 | 0.47 | 0.57 | 0.55 | 14.71 | 0.55 | **0.55** | 1.96 | 14.21 |
| | UDE-MR | 0.45 | 0.44 | 0.96 | 12.19 | 0.55 | 0.46 | 0.40 | 11.54 | 0.53 | 0.57 | 0.31 | 21.74 |
| | UDE-MRVC | 0.44 | 0.44 | 0.91 | 12.07 | 0.53 | **0.41** | **0.34** | 10.08 | 0.52 | 0.56 | 0.31 | 21.28 |
| | UDE-MRVCJ | 0.42 | **0.42** | 0.90 | **11.97** | 0.52 | 0.44 | **0.34** | 10.47 | **0.50** | **0.55** | 0.31 | 20.21 |
| | GARCH | 0.94 | 0.82 | 0.96 | 25.05 | 1.80 | 1.26 | 1.64 | 36.72 | 0.59 | 0.59 | 0.35 | 12.40 |
| | EWMA | 0.73 | 0.69 | 0.78 | 20.77 | 1.43 | 1.10 | 1.36 | 31.40 | 0.62 | 0.65 | 0.35 | 10.87 |
| | FIGARCH | 0.94 | 0.80 | 0.94 | 24.53 | 1.79 | 1.25 | 1.61 | 36.40 | 0.65 | 0.69 | 0.35 | 12.15 |
| | MertonJD | 0.71 | 0.70 | 0.77 | 20.68 | 1.46 | 1.13 | 1.40 | 31.98 | 0.73 | 0.70 | 0.35 | 12.40 |
| | LSTM | 0.39 | 0.49 | 0.95 | 13.71 | 0.61 | 0.65 | 0.64 | 16.90 | 0.91 | 0.83 | 2.19 | 16.90 |
| | GRU | 0.35 | 0.47 | 0.66 | 12.97 | 0.65 | 0.69 | 0.67 | 17.87 | 0.94 | 0.82 | 1.30 | 14.75 |
| | TimesFM | **0.33** | 0.46 | **0.61** | 12.66 | 0.45 | 0.54 | 0.50 | 13.86 | 0.98 | 0.84 | 2.13 | 18.02 |
| | TimesFM FS | **0.33** | 0.45 | 0.64 | 12.52 | 0.42 | 0.52 | 0.46 | 13.17 | 0.99 | 0.85 | 2.21 | 18.46 |
| *Bonds* | | | | | | | | | | | | | |
| AGG | RW | 0.32 | 0.45 | 0.62 | 7.08 | 0.38 | 0.49 | 0.34 | 7.61 | 0.46 | 0.55 | 1.58 | 8.98 |
| | ODE-MR | 0.35 | 0.47 | 0.62 | 7.30 | 0.40 | 0.51 | 0.37 | 7.85 | 0.48 | 0.56 | 1.55 | 9.05 |
| | Vanilla Neural ODE | 0.33 | 0.46 | 0.60 | 7.15 | 0.37 | 0.49 | 0.35 | 7.70 | 0.44 | 0.54 | 1.48 | 8.75 |
| | UDE-MR | **0.30** | **0.30** | 0.52 | **6.35** | **0.35** | **0.35** | **0.26** | **6.95** | 0.43 | 0.43 | 0.65 | 7.95 |
| | UDE-MRVC | 0.31 | 0.31 | 0.55 | 6.90 | 0.37 | 0.37 | 0.33 | 7.70 | 0.43 | 0.39 | 0.52 | 6.55 |
| | UDE-MRVCJ | 0.32 | 0.32 | 0.63 | 7.85 | 0.39 | 0.39 | 0.42 | 8.70 | 0.46 | 0.46 | 0.82 | 8.50 |
| | GARCH | 1.95 | 1.28 | 1.63 | 22.55 | 3.00 | 1.67 | 2.19 | 28.44 | 0.91 | 0.84 | 0.97 | 15.88 |
| | EWMA | 1.30 | 1.01 | 1.19 | 17.32 | 2.09 | 1.37 | 1.64 | 22.81 | 0.51 | 0.61 | 0.70 | 10.99 |
| | FIGARCH | 1.79 | 1.21 | 1.51 | 21.29 | 2.76 | 1.59 | 2.04 | 26.95 | 0.79 | 0.77 | 0.86 | 14.53 |
| | MertonJD | 1.38 | 1.07 | 1.25 | 18.21 | 2.28 | 1.46 | 1.78 | 24.16 | 0.54 | 0.64 | 0.66 | 11.52 |
| | LSTM | 0.50 | 0.57 | 0.59 | 9.14 | 0.91 | 0.84 | 0.76 | 13.07 | 0.24 | 0.40 | 0.67 | 6.74 |
| | GRU | 0.35 | 0.46 | **0.45** | 7.31 | 0.61 | 0.68 | 0.51 | 10.41 | 0.26 | 0.41 | 0.76 | 6.98 |
| | TimesFM | 0.32 | 0.44 | 0.52 | 6.98 | 0.51 | 0.60 | 0.43 | 9.22 | 0.18 | 0.33 | 0.45 | 5.62 |
| | TimesFM FS | **0.30** | 0.43 | 0.51 | 6.78 | 0.48 | 0.58 | 0.40 | 8.82 | **0.16** | **0.32** | **0.41** | **5.42** |

Table 11: Medium Horizon ($h = 10$) Performance: Equities

| Asset | Model | Aggregated | | | | Low Vol | | | | High Vol | | | |
|---|---|---|---|---|---|---|---|---|---|---|---|---|---|
| | | MSE | MAE | QL | sM | MSE | MAE | QL | sM | MSE | MAE | QL | sM |
| *Equities* | | | | | | | | | | | | | |
| DowJones | RW | 0.55 | 0.63 | 0.64 | 12.55 | 1.12 | 1.01 | 1.18 | 19.60 | 0.47 | 0.56 | 0.43 | 11.02 |
| | ODE-MR | 0.34 | 0.46 | 1.09 | 9.03 | 0.38 | 0.49 | 0.67 | 9.33 | 0.52 | 0.58 | 2.85 | 11.76 |
| | Vanilla Neural ODE | 0.33 | 0.45 | 0.93 | 8.93 | 0.39 | 0.51 | **0.50** | 9.63 | 0.45 | 0.53 | 2.32 | 10.85 |
| | UDE-MR | 0.33 | 0.33 | 0.61 | 9.01 | **0.37** | **0.37** | 0.64 | **5.48** | 0.42 | 0.41 | 0.53 | 8.45 |
| | UDE-MRVC | **0.32** | **0.32** | 0.60 | **8.67** | **0.37** | **0.37** | 0.59 | 11.64 | 0.36 | 0.35 | 0.51 | **6.07** |
| | UDE-MRVCJ | 0.34 | 0.34 | **0.58** | **8.67** | 0.38 | 0.38 | 0.57 | 11.21 | **0.33** | **0.32** | 0.55 | 6.25 |
| | GARCH | 1.03 | 0.86 | 1.05 | 18.31 | 1.83 | 1.27 | 1.65 | 25.90 | 0.64 | 0.70 | 0.55 | 14.90 |
| | EWMA | 0.84 | 0.76 | 0.88 | 15.91 | 1.56 | 1.15 | 1.45 | 23.38 | 0.72 | 0.66 | 0.46 | 13.12 |
| | FIGARCH | 0.92 | 0.81 | 0.95 | 17.02 | 1.70 | 1.21 | 1.55 | 24.62 | 0.75 | 0.68 | 0.46 | 13.74 |
| | MertonJD | 0.88 | 0.79 | 0.91 | 16.59 | 1.62 | 1.19 | 1.51 | 24.08 | 0.70 | 0.66 | **0.41** | 13.02 |
| | LSTM | 0.53 | 0.59 | 2.18 | 11.15 | 0.43 | 0.54 | 0.62 | 10.05 | 0.77 | 0.76 | 4.26 | 15.62 |
| | GRU | 0.45 | 0.54 | 1.27 | 10.79 | 0.54 | 0.58 | 0.61 | 11.17 | 0.64 | 0.70 | 3.09 | 14.28 |
| | TimesFM | 0.38 | 0.49 | 1.08 | 9.66 | 0.47 | 0.56 | 0.57 | 10.76 | 0.52 | 0.60 | 2.58 | 12.34 |
| | TimesFM FS | 0.36 | 0.48 | 0.99 | 9.48 | 0.47 | 0.57 | 0.57 | 10.84 | 0.48 | 0.56 | 2.32 | 11.60 |
| FTSE | RW | 0.30 | 0.43 | 0.82 | 8.56 | 0.35 | 0.48 | 0.45 | 9.06 | 0.47 | 0.58 | 2.13 | 11.89 |
| | ODE-MR | 0.50 | 0.61 | 0.60 | 12.16 | 1.00 | 0.95 | 1.08 | 18.70 | 0.42 | 0.52 | **0.18** | 10.82 |
| | Vanilla Neural ODE | 0.35 | 0.47 | 0.96 | 9.41 | 0.42 | 0.54 | 0.55 | 10.35 | 0.46 | 0.54 | 2.12 | 11.38 |
| | UDE-MR | 0.29 | 0.29 | 0.47 | 7.87 | 0.34 | **0.34** | 0.53 | **5.96** | 0.39 | 0.38 | 0.44 | 8.63 |
| | UDE-MRVC | **0.28** | **0.28** | **0.44** | 7.23 | 0.34 | **0.34** | 0.43 | 9.50 | 0.35 | 0.34 | 0.43 | 5.60 |
| | UDE-MRVCJ | 0.30 | 0.30 | **0.44** | **7.16** | 0.35 | 0.35 | 0.42 | 9.24 | **0.32** | **0.31** | 0.43 | **5.40** |
| | GARCH | 0.69 | 0.68 | 0.76 | 14.45 | 1.23 | 1.02 | 1.23 | 20.83 | 0.59 | 0.64 | 0.38 | 11.84 |
| | EWMA | 0.58 | 0.62 | 0.67 | 12.93 | 1.03 | 0.93 | 1.07 | 18.79 | 0.63 | 0.68 | 0.42 | 12.07 |
| | FIGARCH | 0.63 | 0.66 | 0.71 | 13.90 | 1.18 | 1.02 | 1.21 | 20.68 | 0.67 | 0.69 | 0.30 | 12.44 |
| | MertonJD | 0.60 | 0.65 | 0.69 | 13.51 | 1.11 | 0.99 | 1.16 | 19.95 | 0.65 | 0.68 | 0.31 | 12.21 |
| | LSTM | 0.34 | 0.47 | 1.26 | 9.14 | **0.22** | 0.36 | **0.34** | 6.78 | 0.74 | 0.77 | 3.58 | 15.62 |
| | GRU | 0.34 | 0.47 | 1.01 | 9.21 | 0.32 | 0.43 | 0.42 | 8.25 | 0.62 | 0.68 | 2.67 | 13.74 |
| | TimesFM | 0.30 | 0.42 | 0.64 | 8.17 | 0.30 | 0.44 | 0.40 | 8.38 | 0.52 | 0.60 | 2.02 | 11.81 |
| | TimesFM FS | **0.28** | 0.41 | 0.63 | 8.05 | 0.30 | 0.45 | 0.40 | 8.47 | 0.47 | 0.56 | 1.89 | 10.85 |
| NASDAQ | RW | 0.34 | 0.47 | 0.99 | 9.71 | 0.40 | 0.52 | 0.91 | 13.34 | 0.57 | 0.61 | 2.38 | 12.62 |
| | ODE-MR | 0.36 | 0.49 | 0.89 | 10.18 | 0.75 | 0.79 | 0.84 | 15.85 | 0.50 | 0.59 | 0.43 | 12.78 |
| | Vanilla Neural ODE | 0.36 | 0.47 | 1.23 | 9.74 | 0.45 | 0.55 | 0.60 | 10.91 | 0.48 | 0.56 | 2.12 | 11.90 |
| | UDE-MR | 0.33 | 0.33 | 0.65 | 9.29 | 0.42 | 0.44 | 0.64 | 12.91 | 0.43 | 0.41 | 0.51 | 8.92 |
| | UDE-MRVC | **0.32** | **0.32** | 0.63 | 9.07 | **0.39** | **0.39** | **0.59** | **7.16** | 0.39 | 0.37 | 0.50 | **7.49** |
| | UDE-MRVCJ | 0.33 | 0.33 | **0.62** | **9.03** | 0.40 | 0.40 | **0.59** | 12.05 | 0.37 | 0.36 | 0.52 | 7.78 |
| | GARCH | 1.27 | 0.98 | 1.21 | 22.45 | 2.32 | 1.45 | 1.99 | 31.99 | 0.69 | 0.76 | 0.51 | 13.78 |
| | EWMA | 0.92 | 0.81 | 0.94 | 18.11 | 1.74 | 1.25 | 1.60 | 26.80 | 0.72 | 0.78 | 0.41 | 13.32 |
| | FIGARCH | 1.23 | 0.95 | 1.16 | 21.66 | 2.27 | 1.42 | 1.93 | 31.25 | 0.74 | 0.79 | 0.51 | 13.90 |
| | MertonJD | 0.94 | 0.83 | 0.96 | 18.48 | 1.77 | 1.27 | 1.64 | 27.18 | 0.71 | 0.76 | **0.39** | 13.08 |
| | LSTM | 0.53 | 0.57 | 2.43 | 11.63 | 0.44 | 0.54 | 0.70 | 10.68 | 0.90 | 0.83 | 6.20 | 17.40 |
| | GRU | 0.55 | 0.60 | 1.90 | 12.45 | 0.66 | 0.70 | 0.85 | 14.20 | 0.80 | 0.76 | 4.40 | 14.13 |
| | TimesFM | 0.42 | 0.51 | 1.39 | 10.53 | 0.52 | 0.62 | 0.68 | 12.25 | 0.48 | 0.59 | 3.28 | 12.12 |
| | TimesFM FS | 0.40 | 0.50 | 1.28 | 10.31 | 0.52 | 0.62 | 0.67 | 12.22 | 0.46 | 0.58 | 2.99 | 12.51 |
| NIFTY50 | RW | 0.29 | 0.42 | 0.77 | 8.33 | 0.33 | 0.46 | 0.43 | 8.88 | 0.40 | 0.52 | 1.87 | 10.62 |
| | ODE-MR | 0.45 | 0.57 | 0.55 | 11.40 | 0.92 | 0.91 | 1.02 | 17.98 | 0.42 | 0.53 | 0.57 | 11.12 |
| | Vanilla Neural ODE | 0.29 | 0.41 | 0.59 | 8.01 | 0.34 | 0.48 | 0.46 | 9.20 | 0.39 | 0.48 | 1.79 | 9.89 |
| | UDE-MR | 0.30 | 0.30 | 0.44 | 7.36 | 0.33 | 0.33 | 0.48 | 10.52 | 0.39 | 0.35 | 0.45 | 7.31 |
| | UDE-MRVC | 0.28 | 0.28 | 0.42 | 7.23 | 0.32 | 0.32 | 0.44 | 9.69 | 0.37 | 0.31 | 0.45 | 6.01 |
| | UDE-MRVCJ | **0.27** | **0.27** | **0.40** | **7.22** | **0.31** | **0.31** | 0.42 | 8.97 | **0.34** | **0.30** | **0.35** | **5.12** |
| | GARCH | 1.21 | 0.96 | 1.16 | 20.95 | 2.00 | 1.33 | 1.76 | 27.93 | 0.64 | 0.70 | 0.57 | 13.42 |
| | EWMA | 0.85 | 0.78 | 0.88 | 16.66 | 1.49 | 1.14 | 1.42 | 23.44 | 0.69 | 0.74 | 0.38 | 12.45 |
| | FIGARCH | 1.10 | 0.91 | 1.09 | 19.84 | 1.84 | 1.28 | 1.67 | 26.70 | 0.72 | 0.76 | 0.52 | 13.74 |
| | MertonJD | 0.86 | 0.80 | 0.91 | 17.12 | 1.51 | 1.16 | 1.45 | 23.80 | 0.71 | 0.73 | 0.39 | 12.89 |
| | LSTM | 0.39 | 0.49 | 1.37 | 9.82 | 0.42 | 0.49 | 0.53 | **8.74** | 0.75 | 0.80 | 3.49 | 16.29 |
| | GRU | 0.43 | 0.51 | 1.21 | 10.36 | 0.45 | 0.52 | 0.52 | 9.31 | 0.65 | 0.72 | 2.87 | 14.78 |
| | TimesFM | 0.31 | 0.43 | 0.84 | 8.66 | 0.34 | 0.46 | 0.44 | 8.78 | 0.46 | 0.57 | 1.94 | 11.81 |
| | TimesFM FS | 0.29 | 0.42 | 0.76 | 8.39 | 0.34 | 0.46 | 0.44 | 8.89 | 0.42 | 0.53 | 1.71 | 11.06 |
| Russell1000 | RW | 0.32 | 0.44 | 0.85 | 8.78 | 0.34 | 0.46 | 0.44 | 8.90 | 0.46 | 0.58 | 2.05 | 11.81 |
| | ODE-MR | 0.48 | 0.59 | 0.58 | 11.97 | 0.95 | 0.94 | 1.05 | 18.46 | 0.50 | 0.62 | 0.72 | 12.98 |
| | Vanilla Neural ODE | 0.34 | 0.43 | 1.59 | 8.35 | 0.32 | 0.44 | 0.54 | 8.27 | 0.44 | 0.53 | 4.05 | 12.46 |
| | UDE-MR | 0.31 | 0.31 | 0.61 | 8.10 | 0.33 | 0.33 | 0.50 | 10.58 | 0.43 | 0.35 | 0.52 | 7.01 |
| | UDE-MRVC | **0.30** | **0.30** | 0.57 | 8.06 | 0.32 | 0.32 | 0.47 | 10.04 | 0.40 | 0.32 | 0.52 | 6.01 |
| | UDE-MRVCJ | 0.31 | 0.31 | **0.55** | **8.02** | **0.31** | **0.31** | 0.45 | 9.75 | 0.36 | 0.30 | 0.52 | **5.87** |
| | GARCH | 1.63 | 1.13 | 1.45 | 24.89 | 2.58 | 1.53 | 2.14 | 32.72 | 0.67 | 0.74 | 0.61 | 14.04 |
| | EWMA | 1.27 | 0.99 | 1.21 | 21.48 | 2.10 | 1.39 | 1.86 | 29.09 | 0.64 | 0.70 | **0.46** | 12.33 |
| | FIGARCH | 1.65 | 1.13 | 1.45 | 24.97 | 2.58 | 1.53 | 2.12 | 32.67 | 0.70 | 0.76 | 0.63 | 14.25 |
| | MertonJD | 1.30 | 1.02 | 1.26 | 22.05 | 2.15 | 1.42 | 1.92 | 29.68 | 0.68 | 0.74 | **0.46** | 12.82 |
| | LSTM | 0.38 | 0.49 | 1.47 | 9.49 | 0.40 | 0.45 | 0.50 | 8.12 | 0.84 | 0.85 | 4.22 | 17.07 |
| | GRU | 0.38 | 0.48 | 1.14 | 9.41 | 0.43 | 0.46 | **0.41** | **7.74** | 0.67 | 0.75 | 3.07 | 15.12 |
| | TimesFM | 0.32 | 0.44 | 0.93 | 8.74 | 0.35 | 0.47 | **0.41** | 8.30 | 0.54 | 0.64 | 2.43 | 13.10 |
| | TimesFM FS | 0.31 | 0.44 | 0.87 | 8.62 | 0.34 | 0.45 | 0.42 | 8.54 | 0.50 | 0.60 | 2.23 | 12.49 |
| Russell2000 | RW | 0.36 | 0.47 | 1.08 | 9.98 | 0.40 | 0.52 | **0.50** | 10.51 | 0.51 | 0.58 | 2.65 | 12.67 |
| | ODE-MR | 0.36 | 0.46 | 0.65 | 9.32 | 0.59 | 0.70 | 0.70 | 14.23 | 0.46 | 0.54 | 0.51 | 11.60 |
| | Vanilla Neural ODE | 0.39 | 0.48 | 1.89 | 10.09 | **0.36** | 0.49 | 0.66 | 9.88 | 0.49 | 0.56 | 4.65 | 13.20 |
| | UDE-MR | 0.35 | 0.35 | 0.57 | 8.69 | 0.39 | 0.39 | 0.52 | **5.70** | 0.44 | 0.38 | 0.49 | 9.01 |
| | UDE-MRVC | 0.33 | 0.33 | 0.56 | 8.68 | 0.38 | 0.38 | 0.52 | 8.31 | 0.41 | 0.35 | 0.49 | 7.40 |
| | UDE-MRVCJ | **0.32** | **0.32** | 0.54 | 8.52 | 0.37 | **0.37** | 0.51 | 7.95 | 0.38 | 0.33 | 0.46 | **6.74** |
| | GARCH | 1.16 | 0.94 | 1.15 | 21.84 | 2.06 | 1.37 | 1.83 | 30.54 | 0.67 | 0.73 | 0.50 | 13.26 |
| | EWMA | 0.96 | 0.84 | 0.98 | 19.07 | 1.75 | 1.25 | 1.61 | 27.54 | 0.70 | 0.75 | 0.39 | 12.63 |
| | FIGARCH | 1.06 | 0.90 | 1.07 | 20.61 | 1.89 | 1.31 | 1.72 | 29.03 | 0.74 | 0.77 | 0.43 | 13.09 |
| | MertonJD | 0.97 | 0.86 | 1.00 | 19.64 | 1.79 | 1.29 | 1.67 | 28.21 | 0.72 | 0.74 | **0.36** | 12.85 |
| | LSTM | 0.44 | 0.52 | 2.10 | 10.67 | 0.42 | 0.47 | 0.58 | 8.74 | 0.87 | 0.82 | 5.34 | 17.59 |
| | GRU | 0.46 | 0.54 | 1.60 | 11.34 | 0.53 | 0.59 | 0.73 | 12.21 | 0.64 | 0.64 | 3.68 | 14.08 |
| | TimesFM | 0.35 | 0.47 | 1.19 | 9.76 | 0.42 | 0.54 | 0.58 | 10.92 | 0.53 | 0.60 | 2.82 | 13.13 |
| | TimesFM FS | 0.34 | 0.46 | 1.10 | 9.58 | 0.42 | 0.54 | 0.57 | 10.98 | 0.50 | 0.57 | 2.58 | 12.65 |
| SP500 | RW | 0.61 | 0.55 | 1.30 | 14.92 | 1.26 | 1.06 | 1.27 | 20.52 | 0.59 | 0.63 | 3.33 | 12.80 |
| | ODE-MR | 0.63 | 0.66 | 0.71 | 13.23 | 0.84 | 0.59 | **0.65** | 16.03 | 0.49 | 0.56 | **0.28** | 11.53 |
| | Vanilla Neural ODE | 0.60 | 0.60 | 1.94 | 11.90 | 0.65 | 0.65 | 0.80 | 12.42 | 0.71 | 0.67 | 4.61 | 13.68 |
| | UDE-MR | 0.40 | 0.40 | 0.75 | 10.07 | 0.43 | 0.43 | 0.79 | 14.17 | 0.57 | 0.40 | 1.59 | 10.67 |
| | UDE-MRVC | **0.38** | **0.38** | 0.73 | 9.53 | **0.41** | **0.41** | 0.69 | 12.86 | 0.52 | **0.37** | 1.37 | **6.67** |
| | UDE-MRVCJ | 0.39 | 0.39 | **0.69** | **9.49** | 0.44 | 0.44 | 0.67 | 12.54 | 0.55 | 0.43 | 1.31 | 9.10 |
| | GARCH | 1.33 | 1.00 | 1.24 | 21.27 | 2.38 | 1.47 | 2.01 | 30.28 | 0.36 | 0.47 | 0.47 | 11.01 |
| | EWMA | 1.06 | 0.87 | 1.05 | 18.25 | 1.93 | 1.31 | 1.72 | 26.55 | **0.29** | 0.41 | 0.45 | 9.33 |
| | FIGARCH | 1.31 | 0.98 | 1.21 | 20.98 | 2.30 | 1.44 | 1.96 | 29.52 | 0.37 | 0.47 | 0.47 | 11.09 |
| | MertonJD | 1.08 | 0.89 | 1.07 | 18.75 | 1.98 | 1.34 | 1.77 | 27.10 | 0.35 | 0.43 | 0.41 | 9.76 |
| | LSTM | 0.52 | 0.56 | 2.18 | 11.02 | 0.46 | 0.55 | 0.66 | **10.32** | 0.87 | 0.79 | 5.57 | 16.07 |
| | GRU | 0.52 | 0.57 | 1.45 | 11.45 | 0.64 | 0.65 | 0.71 | 12.44 | 0.66 | 0.67 | 3.49 | 13.84 |
| | TimesFM | 0.44 | 0.53 | 1.26 | 10.36 | 0.57 | 0.63 | 0.68 | 11.96 | 0.54 | 0.59 | 2.92 | 12.10 |
| | TimesFM FS | 0.43 | 0.52 | 1.19 | 10.25 | 0.57 | 0.64 | 0.68 | 12.03 | 0.51 | 0.58 | 2.73 | 11.97 |

Table 12: Medium Horizon ($h = 20$) Performance: Crypto & Bonds

| Asset | Model | Aggregated | | | | Low Vol | | | | High Vol | | | |
|---|---|---|---|---|---|---|---|---|---|---|---|---|---|
| | | MSE | MAE | QL | sM | MSE | MAE | QL | sM | MSE | MAE | QL | sM |
| *Crypto* | | | | | | | | | | | | | |
| Bitcoin | RW | 0.57 | 0.60 | 2.30 | 15.56 | 0.65 | 0.65 | 0.77 | 16.08 | 0.86 | 0.76 | 6.53 | 20.65 |
| | ODE-MR | 0.60 | 0.64 | 3.06 | 16.10 | 0.58 | 0.66 | **0.42** | 15.36 | 1.44 | 1.15 | 9.10 | 30.43 |
| | Vanilla Neural ODE | 0.53 | 0.56 | 2.20 | 14.37 | 0.70 | 0.69 | 0.93 | 16.65 | 0.73 | 0.69 | 6.06 | 18.65 |
| | UDE-MR | 0.55 | **0.47** | 1.19 | **8.63** | 0.65 | 0.45 | 0.45 | 9.27 | 0.80 | 0.73 | 3.47 | 20.39 |
| | UDE-MRVC | 0.54 | 0.48 | 1.27 | 12.60 | 0.64 | 0.52 | 0.50 | 9.40 | 0.78 | 0.74 | 3.67 | 20.50 |
| | UDE-MRVCJ | 0.50 | 0.49 | 1.25 | 12.55 | 0.61 | 0.51 | 0.48 | 9.35 | 0.75 | 0.72 | 3.56 | 19.93 |
| | GARCH | 1.48 | 1.03 | 1.31 | 30.04 | 2.69 | 1.55 | 2.17 | 42.33 | **0.36** | 0.46 | 0.45 | 15.46 |
| | EWMA | 0.96 | 0.80 | 0.96 | 22.43 | 1.91 | 1.29 | 1.69 | 33.97 | 0.59 | 0.54 | 0.37 | 10.99 |
| | FIGARCH | 1.43 | 0.99 | 1.25 | 28.69 | 2.59 | 1.49 | 2.07 | 40.50 | **0.36** | **0.45** | 0.45 | 15.01 |
| | MertonJD | 0.89 | 0.78 | **0.90** | 21.41 | 1.83 | 1.28 | 1.66 | 33.31 | 1.21 | 1.16 | **0.34** | **8.90** |
| | LSTM | 0.55 | 0.59 | 1.97 | 15.05 | 0.70 | 0.69 | 0.78 | 16.84 | 0.73 | 0.66 | 5.18 | 17.89 |
| | GRU | 0.48 | 0.55 | 1.08 | 14.16 | 0.85 | 0.79 | 0.85 | 19.26 | 0.45 | 0.52 | 2.40 | 14.64 |
| | TimesFM | 0.45 | 0.53 | 1.39 | 13.56 | 0.62 | 0.65 | 0.67 | 15.44 | 0.62 | 0.64 | 3.66 | 17.73 |
| | TimesFM FS | **0.44** | 0.52 | 1.36 | 13.34 | **0.22** | **0.36** | 0.64 | **8.04** | 0.63 | 0.65 | 3.64 | 18.04 |
| Ethereum | RW | 0.48 | 0.55 | 1.48 | 15.45 | 0.60 | 0.63 | 0.68 | 17.04 | 0.63 | 0.65 | 3.75 | 19.28 |
| | ODE-MR | 0.91 | 0.83 | 5.65 | 21.43 | 0.32 | 0.43 | 0.84 | 10.36 | 1.97 | 1.36 | 15.27 | 37.10 |
| | Vanilla Neural ODE | 0.36 | 0.47 | 0.90 | 13.24 | 0.51 | 0.60 | 0.61 | 15.67 | 0.48 | 0.58 | 2.18 | 17.64 |
| | UDE-MR | 0.46 | 0.44 | 0.94 | 12.70 | 0.59 | **0.42** | 0.36 | **10.24** | 0.60 | 0.60 | 2.67 | 21.39 |
| | UDE-MRVC | 0.45 | 0.44 | 0.99 | 12.55 | 0.58 | 0.47 | 0.42 | 11.83 | 0.57 | 0.57 | 2.86 | 21.72 |
| | UDE-MRVCJ | 0.43 | **0.43** | 0.91 | **12.45** | 0.56 | **0.42** | **0.35** | 10.30 | 0.55 | 0.55 | 2.51 | 19.93 |
| | GARCH | 1.12 | 0.90 | 1.09 | 27.99 | 2.07 | 1.35 | 1.80 | 39.55 | 0.30 | 0.43 | 0.44 | 15.15 |
| | EWMA | 0.80 | 0.73 | 0.84 | 22.06 | 1.59 | 1.17 | 1.48 | 33.30 | **0.18** | **0.33** | 0.34 | 11.34 |
| | FIGARCH | 1.11 | 0.87 | 1.06 | 27.34 | 2.02 | 1.31 | 1.74 | 38.63 | 0.30 | 0.41 | 0.43 | 14.74 |
| | MertonJD | 0.74 | 0.71 | 0.79 | 21.16 | 1.52 | 1.16 | 1.45 | 32.55 | 0.98 | 1.03 | **0.32** | **10.32** |
| | LSTM | 0.39 | 0.49 | 0.94 | 13.70 | 0.63 | 0.63 | 0.65 | 17.26 | 0.45 | 0.55 | 2.13 | 16.62 |
| | GRU | 0.36 | 0.47 | **0.66** | 13.02 | 0.67 | 0.68 | 0.70 | 18.18 | 0.32 | 0.47 | 1.28 | 14.49 |
| | TimesFM | 0.35 | 0.47 | 0.85 | 12.88 | 0.51 | 0.56 | 0.55 | 14.78 | 0.45 | 0.57 | 2.03 | 17.30 |
| | TimesFM FS | **0.34** | 0.46 | 0.85 | 12.81 | **0.26** | 0.54 | 0.53 | **10.24** | 0.46 | 0.58 | 2.07 | 17.55 |
| *Bonds* | | | | | | | | | | | | | |
| AGG | RW | 0.30 | 0.43 | 0.59 | 6.80 | 0.36 | 0.48 | 0.32 | 7.37 | 0.43 | 0.53 | 1.57 | 8.61 |
| | ODE-MR | 0.37 | 0.48 | 0.64 | 7.45 | 0.43 | 0.53 | 0.39 | 7.95 | 0.50 | 0.57 | 1.57 | 9.10 |
| | Vanilla Neural ODE | 0.35 | 0.47 | 0.62 | 7.30 | 0.40 | 0.51 | 0.37 | 7.80 | 0.48 | 0.56 | 1.52 | 8.95 |
| | UDE-MR | **0.28** | **0.28** | 0.54 | **6.50** | **0.33** | **0.33** | **0.28** | **7.15** | 0.41 | 0.41 | 0.72 | 8.05 |
| | UDE-MRVC | 0.29 | 0.29 | 0.58 | 7.05 | 0.35 | 0.35 | 0.35 | 7.75 | 0.23 | 0.38 | 0.57 | 6.52 |
| | UDE-MRVCJ | 0.30 | 0.30 | 0.67 | 8.10 | 0.37 | 0.37 | 0.46 | 8.95 | **0.22** | **0.37** | **0.53** | **6.40** |
| | GARCH | 2.41 | 1.42 | 1.89 | 25.40 | 3.56 | 1.81 | 2.44 | 31.15 | 1.24 | 0.97 | 1.22 | 18.70 |
| | EWMA | 1.42 | 1.06 | 1.28 | 18.30 | 2.28 | 1.44 | 1.75 | 23.95 | 0.64 | 0.67 | 0.83 | 12.23 |
| | FIGARCH | 2.17 | 1.33 | 1.73 | 23.67 | 3.18 | 1.70 | 2.24 | 29.06 | 1.03 | 0.86 | 1.05 | 16.41 |
| | MertonJD | 1.43 | 1.08 | 1.28 | 18.47 | 2.33 | 1.47 | 1.80 | 24.40 | 0.57 | 0.65 | 0.71 | 11.76 |
| | LSTM | 0.51 | 0.58 | 0.61 | 9.23 | 0.91 | 0.84 | 0.75 | 13.01 | 0.26 | 0.41 | 0.75 | 6.96 |
| | GRU | 0.36 | 0.47 | **0.47** | 7.42 | 0.62 | 0.69 | 0.51 | 10.58 | 0.28 | 0.43 | 0.83 | 7.34 |
| | TimesFM | 0.35 | 0.47 | 0.55 | 7.41 | 0.57 | 0.64 | 0.48 | 9.86 | 0.39 | 0.39 | 0.58 | 7.20 |
| | TimesFM FS | 0.34 | 0.46 | 0.56 | 7.23 | 0.54 | 0.62 | 0.45 | 9.49 | 0.43 | 0.43 | 0.92 | 8.70 |

Table 13: Medium Horizon ($h = 20$) Performance: Equities

| Asset | Model | Aggregated | | | | Low Vol | | | | High Vol | | | |
|---|---|---|---|---|---|---|---|---|---|---|---|---|---|
| | | MSE | MAE | QL | sM | MSE | MAE | QL | sM | MSE | MAE | QL | sM |
| *Equities* | | | | | | | | | | | | | |
| DowJones | RW | 0.38 | 0.50 | 1.26 | 9.75 | 1.09 | 0.99 | 1.15 | 19.32 | 0.59 | 0.64 | 3.26 | 13.01 |
| | ODE-MR | 0.55 | 0.63 | 0.65 | 12.56 | 0.43 | 0.52 | **0.52** | 9.81 | 0.52 | 0.62 | **0.27** | 13.05 |
| | Vanilla Neural ODE | 0.48 | 0.55 | 1.48 | 10.90 | 0.53 | 0.60 | 0.67 | 11.43 | 0.63 | 0.65 | 3.68 | 13.36 |
| | UDE-MR | 0.37 | 0.37 | 0.66 | 9.22 | **0.38** | 0.38 | 0.63 | **6.10** | 0.54 | 0.45 | 1.32 | 9.39 |
| | UDE-MRVC | **0.35** | **0.35** | 0.64 | 8.83 | **0.38** | **0.38** | 0.58 | 11.35 | 0.52 | **0.36** | 1.49 | 7.30 |
| | UDE-MRVCJ | 0.38 | 0.38 | 0.67 | 8.85 | 0.40 | 0.40 | 0.55 | 10.95 | 0.41 | 0.41 | 1.36 | **6.51** |
| | GARCH | 1.16 | 0.91 | 1.12 | 19.64 | 2.03 | 1.33 | 1.77 | 27.63 | 0.34 | 0.46 | 0.50 | 10.79 |
| | EWMA | 0.89 | 0.78 | 0.91 | 16.49 | 1.61 | 1.18 | 1.50 | 24.07 | **0.24** | 0.38 | 0.42 | 8.62 |
| | FIGARCH | 1.02 | 0.85 | 1.01 | 18.08 | 1.78 | 1.24 | 1.61 | 25.51 | 0.30 | 0.43 | 0.45 | 9.93 |
| | MertonJD | 0.88 | 0.79 | 0.92 | 16.70 | 1.59 | 1.18 | 1.49 | 23.94 | 0.39 | 0.51 | 0.42 | 9.31 |
| | LSTM | 0.58 | 0.63 | 2.48 | 11.87 | 0.46 | 0.56 | 0.67 | 10.71 | 0.85 | 0.80 | 5.02 | 16.78 |
| | GRU | 0.46 | 0.54 | 1.36 | 10.86 | 0.51 | 0.56 | 0.59 | 10.80 | 0.65 | 0.68 | 3.34 | 14.14 |
| | TimesFM | 0.40 | 0.50 | 1.27 | 9.98 | 0.45 | 0.54 | 0.56 | 10.40 | 0.59 | 0.64 | 3.12 | 13.13 |
| | TimesFM FS | 0.39 | 0.50 | 1.19 | 9.82 | 0.45 | 0.55 | 0.56 | 10.47 | 0.55 | 0.61 | 2.88 | 12.50 |
| FTSE | RW | 0.33 | 0.46 | 0.86 | 8.99 | 0.41 | 0.51 | 0.50 | 9.72 | 0.49 | 0.59 | 2.20 | 12.14 |
| | ODE-MR | 0.51 | 0.61 | 0.61 | 12.30 | 1.01 | 0.96 | 1.10 | 18.85 | 0.46 | 0.64 | **0.18** | 13.13 |
| | Vanilla Neural ODE | 0.60 | 0.61 | 2.07 | 12.31 | 0.67 | 0.67 | 0.84 | 13.30 | 0.79 | 0.69 | 5.44 | 14.10 |
| | UDE-MR | 0.32 | 0.32 | 0.49 | 8.12 | 0.39 | 0.39 | 0.56 | 11.39 | 0.45 | 0.32 | 0.94 | 5.30 |
| | UDE-MRVC | 0.30 | **0.30** | **0.44** | 7.30 | 0.40 | 0.40 | 0.45 | 9.78 | 0.44 | **0.31** | 1.04 | **5.23** |
| | UDE-MRVCJ | 0.33 | 0.33 | **0.44** | **7.22** | 0.42 | 0.42 | 0.43 | 9.50 | 0.47 | 0.44 | 0.99 | 9.18 |
| | GARCH | 0.79 | 0.73 | 0.83 | 15.62 | 1.38 | 1.08 | 1.33 | 22.20 | 0.24 | 0.39 | 0.40 | 8.87 |
| | EWMA | 0.65 | 0.65 | 0.73 | 13.75 | 1.16 | 0.99 | 1.17 | 20.14 | 0.21 | 0.35 | 0.42 | 7.88 |
| | FIGARCH | 0.72 | 0.71 | 0.78 | 15.02 | 1.30 | 1.08 | 1.31 | 21.90 | **0.19** | 0.35 | 0.32 | 7.92 |
| | MertonJD | 0.62 | 0.66 | 0.71 | 13.83 | 1.16 | 1.02 | 1.20 | 20.50 | 0.39 | 0.57 | 0.29 | 6.93 |
| | LSTM | 0.34 | 0.47 | 1.24 | 9.14 | **0.23** | 0.37 | 0.36 | **6.94** | 0.73 | 0.78 | 3.59 | 15.64 |
| | GRU | 0.34 | 0.47 | 1.00 | 9.22 | 0.33 | 0.45 | 0.44 | 8.54 | 0.60 | 0.69 | 2.70 | 13.99 |
| | TimesFM | 0.29 | 0.43 | 0.85 | 8.39 | 0.30 | 0.45 | 0.41 | 8.44 | 0.51 | 0.62 | 2.26 | 12.53 |
| | TimesFM FS | **0.28** | 0.42 | 0.61 | 8.28 | 0.31 | 0.45 | 0.41 | 8.54 | 0.49 | 0.59 | 2.14 | 12.21 |
| NASDAQ | RW | 0.38 | 0.49 | 1.26 | 10.04 | 0.41 | 0.52 | **0.52** | 10.47 | 0.63 | 0.67 | 3.43 | 14.33 |
| | ODE-MR | 0.37 | 0.50 | **0.51** | 10.34 | 0.73 | 0.79 | 0.83 | 15.75 | 0.56 | 0.65 | 0.50 | 14.19 |
| | Vanilla Neural ODE | 0.64 | 0.58 | 3.53 | 11.80 | 0.69 | 0.68 | 1.00 | 13.40 | 0.92 | 0.66 | 9.90 | 13.85 |
| | UDE-MR | 0.39 | 0.39 | 0.72 | 9.60 | 0.42 | 0.42 | 0.65 | 13.05 | 0.58 | 0.44 | 1.56 | 8.31 |
| | UDE-MRVC | **0.36** | **0.36** | 0.69 | 9.26 | **0.39** | **0.39** | 0.59 | 12.12 | 0.55 | **0.41** | 1.63 | **8.01** |
| | UDE-MRVCJ | 0.37 | 0.37 | 0.71 | 9.29 | 0.41 | 0.41 | 0.59 | 12.08 | 0.60 | 0.42 | 1.57 | 8.71 |
| | GARCH | 1.47 | 1.07 | 1.34 | 24.72 | 2.52 | 1.51 | 2.09 | 33.62 | 0.50 | 0.59 | 0.59 | 14.88 |
| | EWMA | 0.98 | 0.84 | 0.99 | 18.83 | 1.81 | 1.27 | 1.65 | 27.47 | **0.31** | 0.43 | 0.45 | 10.51 |
| | FIGARCH | 1.46 | 1.03 | 1.30 | 24.00 | 2.48 | 1.47 | 2.03 | 32.82 | 0.55 | 0.56 | 0.59 | 14.54 |
| | MertonJD | 0.95 | 0.84 | 0.98 | 18.66 | 1.76 | 1.26 | 1.63 | 27.10 | 0.43 | 0.56 | **0.43** | 10.63 |
| | LSTM | 0.53 | 0.57 | 2.56 | 11.70 | 0.43 | 0.53 | 0.71 | 10.52 | 0.98 | 0.84 | 6.69 | 17.92 |
| | GRU | 0.55 | 0.61 | 1.98 | 12.54 | 0.64 | 0.69 | 0.85 | 14.02 | 0.75 | 0.69 | 4.74 | 14.82 |
| | TimesFM | 0.45 | 0.53 | 1.64 | 10.98 | 0.52 | 0.61 | 0.71 | 12.19 | 0.65 | 0.64 | 4.05 | 14.00 |
| | TimesFM FS | 0.43 | 0.52 | 1.50 | 10.73 | 0.51 | 0.61 | 0.69 | 12.18 | 0.60 | 0.62 | 3.70 | 13.54 |
| NIFTY50 | RW | 0.31 | 0.44 | 0.86 | 8.73 | 0.36 | 0.47 | 0.44 | 9.11 | 0.46 | 0.56 | 2.12 | 11.51 |
| | ODE-MR | 0.45 | 0.57 | 0.55 | 11.41 | 0.92 | 0.91 | 1.02 | 17.97 | 0.45 | 0.60 | **0.19** | 12.44 |
| | Vanilla Neural ODE | 0.40 | 0.48 | 1.67 | 9.36 | 0.46 | 0.56 | 0.66 | 10.76 | 0.56 | 0.56 | 4.12 | 11.24 |
| | UDE-MR | 0.31 | 0.31 | **0.43** | 7.51 | 0.34 | **0.34** | 0.49 | **6.42** | 0.46 | 0.37 | 0.82 | 7.14 |
| | UDE-MRVC | 0.29 | 0.29 | 0.46 | 7.41 | 0.35 | 0.35 | 0.45 | 9.84 | 0.41 | 0.41 | 1.06 | 9.11 |
| | UDE-MRVCJ | **0.28** | **0.28** | 0.47 | **7.35** | 0.36 | 0.36 | 0.43 | 9.49 | 0.40 | 0.28 | 0.99 | **5.14** |
| | GARCH | 1.41 | 1.03 | 1.30 | 23.01 | 2.18 | 1.38 | 1.86 | 29.36 | 0.64 | 0.63 | 0.69 | 15.42 |
| | EWMA | 0.92 | 0.81 | 0.93 | 17.46 | 1.55 | 1.17 | 1.47 | 24.12 | 0.30 | 0.42 | 0.39 | 9.74 |
| | FIGARCH | 1.28 | 0.97 | 1.20 | 21.52 | 2.00 | 1.32 | 1.75 | 27.90 | 0.55 | 0.58 | 0.61 | 13.99 |
| | MertonJD | 0.88 | 0.81 | 0.93 | 17.36 | 1.53 | 1.17 | 1.47 | 24.02 | 0.29 | 0.44 | 0.42 | 10.17 |
| | LSTM | 0.40 | 0.50 | 1.40 | 9.86 | **0.27** | 0.37 | **0.38** | 7.06 | 0.78 | 0.83 | 3.61 | 16.85 |
| | GRU | 0.43 | 0.52 | 1.25 | 10.44 | 0.42 | 0.48 | 0.51 | 9.40 | 0.68 | 0.74 | 3.01 | 15.22 |
| | TimesFM | 0.33 | 0.45 | 0.97 | 9.03 | 0.34 | 0.45 | 0.44 | 8.73 | **0.09** | **0.24** | 2.32 | **5.14** |
| | TimesFM FS | 0.32 | 0.44 | 0.89 | 8.81 | 0.34 | 0.46 | 0.44 | 8.84 | 0.48 | 0.59 | 2.12 | 12.19 |
| Russell1000 | RW | 0.35 | 0.46 | 0.95 | 9.11 | 0.39 | 0.50 | 0.48 | 9.71 | 0.50 | 0.59 | 2.25 | 12.01 |
| | ODE-MR | 0.48 | 0.59 | 0.58 | 11.92 | 0.95 | 0.94 | 1.06 | 18.55 | 0.53 | 0.71 | **0.14** | 14.33 |
| | Vanilla Neural ODE | 0.64 | 0.54 | 3.31 | 10.01 | 0.52 | 0.51 | 0.81 | 9.26 | 1.04 | 0.75 | 9.70 | 14.36 |
| | UDE-MR | 0.35 | 0.35 | **0.50** | 7.99 | 0.38 | 0.38 | 0.50 | 10.53 | 0.47 | **0.33** | 1.00 | **5.01** |
| | UDE-MRVC | 0.32 | 0.34 | 0.53 | 7.15 | 0.36 | 0.36 | 0.47 | **5.49** | 0.44 | 0.44 | 1.29 | 9.85 |
| | UDE-MRVCJ | **0.29** | **0.31** | 0.56 | **6.91** | 0.39 | 0.39 | 0.45 | 9.77 | 0.46 | 0.46 | 1.14 | 9.48 |
| | GARCH | 1.91 | 1.21 | 1.60 | 27.14 | 2.89 | 1.60 | 2.26 | 34.49 | 0.80 | 0.69 | 0.77 | 16.75 |
| | EWMA | 1.36 | 1.03 | 1.28 | 22.40 | 2.20 | 1.42 | 1.92 | 29.77 | 0.48 | 0.54 | 0.54 | 12.74 |
| | FIGARCH | 1.97 | 1.21 | 1.62 | 27.51 | 2.93 | 1.60 | 2.27 | 34.75 | 0.85 | 0.69 | 0.79 | 17.02 |
| | MertonJD | 1.31 | 1.03 | 1.27 | 22.22 | 2.13 | 1.41 | 1.90 | 29.45 | **0.42** | 0.53 | 0.50 | 12.31 |
| | LSTM | 0.37 | 0.48 | 1.37 | 9.31 | **0.20** | 0.30 | 0.28 | 5.67 | 0.79 | 0.83 | 3.81 | 16.67 |
| | GRU | 0.37 | 0.47 | 1.07 | 9.23 | 0.22 | 0.32 | 0.37 | 7.48 | 0.65 | 0.74 | 2.81 | 15.11 |
| | TimesFM | 0.32 | 0.45 | 0.93 | 8.74 | 0.30 | 0.42 | 0.38 | 7.85 | 0.55 | 0.66 | 2.39 | 13.38 |
| | TimesFM FS | 0.32 | 0.44 | 0.84 | 8.65 | 0.30 | 0.43 | 0.38 | 8.06 | 0.52 | 0.63 | 2.24 | 12.90 |
| Russell2000 | RW | 0.38 | 0.50 | 1.29 | 10.43 | 0.42 | 0.54 | 0.53 | 11.04 | 0.59 | 0.64 | 3.29 | 13.94 |
| | ODE-MR | **0.30** | 0.45 | **0.47** | 9.50 | 0.59 | 0.71 | 0.70 | 14.40 | 0.49 | 0.61 | 0.61 | 13.44 |
| | Vanilla Neural ODE | 0.81 | 0.61 | 9.52 | 12.37 | 0.64 | 0.58 | 2.72 | 11.36 | 1.49 | 0.86 | 26.08 | 17.82 |
| | UDE-MR | 0.38 | 0.38 | 0.61 | 8.82 | 0.41 | 0.41 | **0.52** | 11.52 | 0.57 | 0.47 | 1.40 | 10.75 |
| | UDE-MRVC | 0.36 | 0.36 | 0.62 | 8.91 | 0.40 | **0.40** | 0.53 | **5.89** | 0.55 | 0.38 | 1.43 | **8.16** |
| | UDE-MRVCJ | 0.35 | **0.35** | 0.63 | **6.32** | 0.43 | 0.43 | 0.53 | 5.25 | 0.51 | 0.48 | 1.42 | 10.80 |
| | GARCH | 1.27 | 0.98 | 1.22 | 22.98 | 2.18 | 1.41 | 1.90 | 31.47 | 0.46 | 0.55 | 0.58 | 14.16 |
| | EWMA | 1.03 | 0.87 | 1.05 | 20.01 | 1.87 | 1.29 | 1.69 | 28.58 | 0.32 | 0.45 | 0.50 | 11.20 |
| | FIGARCH | 1.18 | 0.94 | 1.13 | 21.82 | 2.06 | 1.36 | 1.82 | 30.43 | 0.37 | 0.49 | 0.47 | 12.42 |
| | MertonJD | 0.99 | 0.87 | 1.02 | 19.86 | 1.81 | 1.30 | 1.69 | 28.43 | 0.41 | 0.52 | **0.42** | 10.84 |
| | LSTM | 0.46 | 0.52 | 2.25 | 10.88 | **0.31** | 0.44 | 0.56 | 8.70 | **0.19** | **0.35** | 5.99 | **8.16** |
| | GRU | 0.47 | 0.55 | 1.73 | 11.58 | 0.53 | 0.59 | 0.72 | 12.21 | 0.70 | 0.68 | 4.19 | 14.90 |
| | TimesFM | 0.39 | 0.49 | 1.44 | 10.26 | 0.42 | 0.54 | 0.59 | 10.88 | 0.64 | 0.66 | 3.61 | 14.58 |
| | TimesFM FS | 0.37 | 0.48 | 1.33 | 10.08 | 0.42 | 0.54 | 0.58 | 10.96 | 0.58 | 0.62 | 3.33 | 13.78 |
| SP500 | RW | 0.46 | 0.54 | 1.74 | 10.53 | 0.43 | 0.54 | **0.56** | 10.13 | 0.75 | 0.72 | 4.77 | 14.60 |
| | ODE-MR | 0.63 | 0.66 | **0.72** | 13.28 | 1.24 | 1.05 | 1.26 | 20.42 | 0.62 | 0.67 | **0.33** | 14.01 |
| | Vanilla Neural ODE | 1.41 | 0.86 | 5.70 | 17.47 | 1.37 | 0.88 | 1.42 | 17.32 | 1.62 | 0.94 | 16.22 | 18.76 |
| | UDE-MR | 0.45 | 0.45 | 0.84 | 10.43 | 0.41 | 0.41 | 0.79 | 14.22 | 0.72 | 0.50 | 1.70 | 9.97 |
| | UDE-MRVC | **0.41** | **0.41** | 0.82 | 9.78 | **0.40** | **0.40** | 0.69 | **5.64** | 0.64 | 0.45 | 1.88 | **7.23** |
| | UDE-MRVCJ | 0.44 | 0.44 | 0.78 | **9.76** | 0.43 | 0.43 | 0.67 | 12.46 | 0.68 | 0.48 | 1.69 | **7.23** |
| | GARCH | 1.52 | 1.06 | 1.35 | 23.05 | 2.61 | 1.53 | 2.13 | 31.91 | 0.46 | 0.54 | 0.54 | 12.82 |
| | EWMA | 1.12 | 0.89 | 1.09 | 18.87 | 2.02 | 1.34 | 1.78 | 27.38 | 0.30 | 0.42 | 0.44 | 9.73 |
| | FIGARCH | 1.51 | 1.05 | 1.33 | 22.75 | 2.56 | 1.50 | 2.09 | 31.30 | 0.49 | 0.54 | 0.54 | 12.93 |
| | MertonJD | 1.09 | 0.90 | 1.08 | 18.86 | 1.96 | 1.33 | 1.76 | 27.03 | 0.47 | 0.56 | 0.43 | 10.13 |
| | LSTM | 0.53 | 0.57 | 2.41 | 11.17 | 0.43 | 0.53 | 0.64 | 10.00 | 0.96 | 0.84 | 6.51 | 17.06 |
| | GRU | 0.52 | 0.58 | 1.59 | 11.58 | 0.60 | 0.63 | 0.68 | 12.03 | 0.72 | 0.70 | 4.05 | 14.65 |
| | TimesFM | 0.46 | 0.54 | 1.51 | 10.73 | 0.54 | 0.61 | 0.66 | 11.59 | 0.65 | 0.65 | 3.84 | 13.51 |
| | TimesFM FS | 0.45 | 0.54 | 1.42 | 10.60 | 0.54 | 0.62 | 0.66 | 11.70 | 0.61 | 0.63 | 3.57 | 13.16 |

