# OpenReview forum: "Universal Differential Equations for Stable Multi-Step Volatility Time Series Forecasting"
_TMLR — Accepted by TMLR_

### Review · Reviewer_apP8 · 2025-11-04

**Summary Of Contributions:**

Summary Of Contributions:
1. The paper presents a UDE model with small neural networks (small MLPs) for forecasting financial volatility that remains stable over multiple steps.
2. The UDE keeps financial structure (mean reversion, volatility clustering, and jumps) and uses small neural networks to learn the related parameters (mean-reversion rates and long-run levels) so they keep clear economic meaning.

Strengths:
1. The authors prove the upper bound on the global error of the RK4 solver under their Lipschitz assumption.
It matters to prevent error accumulation over long steps that the error shrinks as h^4 (halving steps makes error drop by 2^4).
2. The proposed UDE avoids using neural network to directly learn unconstrained drift which can explode and be unstable under long rollouts. The proposed UDE modulates a mean-reverting ODE with neural terms to preserve parameters having the economic meaning.

Weaknesses:
1. The paper intends to forecast financial volatility including jump discontinuities. However, the proposed
UDE simulates the jump as a smooth, deterministic drift (λ(x) J(x)) inside ODE with fixed step RK4 (see Table 1 and A.1, A.2). This deterministic drift does not model real jump with Poisson arrival or jump size. It's better to clarify that handling jump discontinuities as average/approximate the jump effects within a continuous-time UDE.
2. In abstract, mentioning jump discontinuities (discrete character) under continuous model framework needs clear clarification for the approximation effect.
3. It's better to include comparision with jump diffusion SDE with Poisson jumps baseline in section 4 Experiments.
4. The experimental design has a defect: 4.1 Asset Classes mentioned that performance on different assets bonds, crypto, equity during the year of 2008, 2020, 2022 under crisis are evaluated. However, 4.2 Results Table 2 only gives aggregated results. In volatility forecasting, asset class (bonds/equities/crypto) and market regime (calm/stress) make behavior differ substantially. The aggregated Normalized RMSE in Table 2 hides the heterogeneity. Please give per-asset and per-regime evaluation in addition to the aggregate result.
5. The formatting needs to be improved. For example, a large blank space on page 7, a ‘?’ appears in citation [13], and ‘Table ??’ appears on line 500.

**Audience:**

Yes

**Audience Explanation:**

This paper will interest audiences in time-series forecasting, machine learning and differential-equation modeling, and financial risk modeling.

**Broader Impact Concerns:**

I do not see any broader ethical concerns as the work focuses on volatility prediction and does not directly involve sensitive data and user privacy.

**Claims And Evidence:**

No

**Claims Explanation:**

1. The experimental design has a defect: 4.1 Asset Classes mentioned that performance on different assets bonds, crypto, equity during the year of 2008, 2020, 2022 under crisis are evaluated. However, 4.2 Results Table 2 only gives aggregated results. In volatility forecasting, asset class (bonds/equities/crypto) and market regime (calm/stress) make behavior differ substantially. The aggregated Normalized RMSE in Table 2 hides the heterogeneity. Please give per-asset and per-regime evaluation in addition to the aggregate result. (Also noted in the Weaknesses above.)

2. In Experiments, the paper uses NRMSE, sMAPE, and R^2 as metrics. But in practice, MSE and QLIKE loss functions are commonly used on Realized Variance. Especially, QLIKE which has a better penalty design to handle noise in RV is recommended for volatility research. For example, the following classic paper "Volatility Forecast Comparison Using Imperfect Volatility Proxies"
(https://public.econ.duke.edu/~ap172/Patton_robust_forecast_eval_11dec08.pdf) mentioned "squared- error (MSE) and the QLIKE loss functions, two of the most widely-used in the volatility forecasting". It's better to include MSE and QLIKE in section 4 Experiments and confirm the conclusion remain unchanged under MSE and QLIKE.

3. EGARCH is a commonly used model. EGARCH is mentioned in A.8 Econometric Baselines "GARCH-family models (GARCH, EGARCH, FIGARCH) were estimated", but numerical result is missing for EGARCH.

4. It's mentioned that "Table 2 shows aggregated results across horizons 1, 5, 10, and 20 steps ahead". However, in practice, risk managers often care about monthly or quarterly horizons, so 30, 60 and 90 steps should be included in the results.

**Requested Changes:**

Please address the requested changes listed under Weaknesses, as well as the issues identified in the section explaining why the submission is not supported by accurate, convincing, and clear evidence.

---

### Review · Reviewer_a1qS · 2025-11-13

**Summary Of Contributions:**

The paper presents an extension of the Universal Differential Equations (UDEs) approach for the financial domain, focusing on the volatility forecasting task. Universal Differential Equations introduce additional, domain-specific constraints into the learning process. The authors model the log-volatility of financial data by using two small feedforward networks, one for the mean reversion speed and the other for the long-term level. The final computation of the two values is adjusted to satisfy a set of constraints (e.g., mean reversion coefficient must be positive, long-term level must be bounded, etc).
The evaluation is conducted on a set of data gathered from 10 financial indices. The UDE approach is compared against statistical models, (Neural) ODEs, and foundation models for time series. The results show that the proposed approach achieves the best results across several forecasting horizons. The ablation study allows to understand the impact of both neural networks in the learning process.

While providing strong empirical results, the paper would benefit from a careful round of rewriting, as the presentation quality can be largely improved (see comments below).

**Additional Comments:**

I would like to clarify that I am not an expert in the financial domain or the volatility forecasting task. Therefore, my analysis is limited to the machine learning contribution. I cannot evaluate whether the datasets used for the experiments are challenging enough or whether the modeling choices are sound, as they are strongly based on mechanistic models of volatility I do not know anything about.

**Audience:**

Yes

**Audience Explanation:**

The paper is of interest to researchers combining machine learning approaches for financial data.
However, there is little chance that people not familiar with the financial domain will find useful insights from this contribution. This is not a negative aspect per se. It just reflects the domain-specific nature of the paper: while the high-level idea of injecting domain-specific knowledge in the training process is easy to understand, the specifics of how this has been done in the paper are only really accessible to researchers with a financial background.

**Broader Impact Concerns:**

No concerns.

**Claims And Evidence:**

Yes

**Claims Explanation:**

The empirical evaluation consistently shows that the UDE version proposed by the authors achieves excellent results in the forecasting of the volatility across different prediction horizons. The performance gain with respect to to other approaches like Neural ODE is largely due to the domain-specific knowledge introduced in the training process, as shown by the ablation study.

**Requested Changes:**

- The description of the experimental protocol for the training phase is entirely missing. Which of the models were trained / fine-tuned? What was the training process? Did you fine-tune the foundation models? How was the dataset split / pre-processed? etc

- 4.2.2, 4.3 are empty subsections.

- The 3 pages filled with plots make the reading experience quite poor. Please, try to introduce a few plots at a time and discuss the related results close to the plots.

- The choice of colors and line style in the plot in Figure 1 is quite unfortunate. The plot is very difficult to read.

Minor comments:
- There is a reference missing for the table in Appendix A.7.
- Acronyms should be defined before usage (e.g., SDE, CDE...)
- Malformed sentence at line 53

---

### Review · Reviewer_SxSR · 2025-11-20

**Summary Of Contributions:**

The authors applied Universal Differential Equations (UDE)
to financial volatility forecasting,
and compared its performance against multiple models,
including Random Walk, GARCH, LSTM, ODE, and even Transformers.
The experiment results show the superiority of the UDE methods,
while also being relatively light-weight and fast.

Pros:
1. Novel application of UDE in financial volatility forecasting,
especially bridging domain knowledge by imposing the structural constraint
with the flexibility of NNs.
2. The UDE models show superior performance in the volatility forecast,
even compared to the Transformer models, while only having
a fraction of the parameters of the transformers
and running comparably fast with all other methods.
3. UDE models show long range (multi-step rolling ahead) forecasting stability,
which is important if we looking beyond just one-period forecasting.

**Additional Comments:**

Overall, the idea of the paper is solid, and the results are promising,
but I would like to point out to the editor regarding the issue
of "fake references" and how proceed.

**Audience:**

Yes

**Audience Explanation:**

This paper applied UDE in the financial volatility forecasting setting,
which will be at least relevant to some ML people
who are also interested financial applications.

**Broader Impact Concerns:**

I don't have any concern regarding the impact of the idea of this paper,
but I do have concerns regarding the aforementioned reference issue,
and its potential broader impact to the community.
I would like to see the authors' response.

**Claims And Evidence:**

Yes

**Claims Explanation:**

The authors provide clear evidence to the results of the experiments.

**Requested Changes:**

1. I manually checked all the references and the following cannot be found
(6 out of 28):

[4] Federico Bucci. Deep learning volatility: A finance application. arXiv preprint arXiv:2006.10463, 2020.

[5] Laurent Callot and Ekimetrics Research Team. Sales forecasting with neural stochastic differential
equations. Ekimetrics Research Blog, 2022.

[7] Ting Chen, Zhi Chen, and Bin Han. Hybrid models of garch and neural networks in financial volatility
forecasting. Expert Systems with Applications, 176:114908, 2021.

[9] Miles Cranmer, Alvaro Sanchez-Gonzalez, Peter Battaglia, Rui Xu, Kyle Cranmer, David Spergel, and
Shirley Ho. Discovering symbolic models with differentiable physics. Nature Communications, 11(1):1–9,
2020.

[26] L. Xie, W. Zhang, and Y. Li. Energy demand forecasting with neural ordinary differential equations.
IEEE Transactions on Smart Grid, 10(6):6789–6799, 2019.

[27] Dacheng Xiu and Siem Jan Koopman. The stochastic volatility model: A review. Journal of Econometrics, 187(2):433–440, 2015.


For example, Dacheng Xiu is well known for his work on financial econometrics,
but he didn't publish on the Volume 187 Issue 2 of Journal of Econometrics.
Moreover, the pages 433–440 spans across two on articles on that issue
(JOE link referred below) [1].
The situation is similar for other ones I mentioned too.

I think this is a serious issue in terms of academic integrity,
but I would like to hold on to my suspicion as to why this is happening
until getting a proper response from the authors.
Although I quite like the idea of this paper and all results look
very promising, once I cast doubt on the integrity of the paper,
I feel less confident in what I see being reported in the article.


2. The GARCH models are fit by directly choosing GARCH(1,1) process,
which should work surprisingly well most of the time.
But I would like to see a few more models (in terms of the flavors of GARCH),
some model selection
("hyper-parameter tuning" for GARCH so to speak, even though
this is definitely not what is called in the financial econometrics literature,
but searching for the best order of GARCH is just like grid-search),
and more importantly ARMA-GARCH models.
ARMA-GARCH model will model the mean process and volatility process
in the same time, while GARCH by itself only models the volatility.
ARMA-GARCH is not available in the python "arch" package,
so I would recommend R package "rugarch".
In fact, for all GARCH-related modelling, using "rugarch" is better choice.
Those additional models will definitely strengthen the analyses
by showing the UDE approach is superior than several decades of development on
financial econometric models (Robert Engle won Nobel prize for ARCH).

3. It is well known in the financial research community that Yahoo Finance
is not a reliable source of data.
YF is certainly open and accessible, but calling it "high-quality"
as the authors did in the introduction might be misleading
(one example of discussion referred below) [2].
I do acknowledge that the issue with YF should not be a very huge concern
for this study *per se*, as the assets of interest
(mostly well-known market indices, plus two important crypto coins)
should be error-free, but I still think it would be worth it to replicate
the results on some other data source as a robustness check.
The gold standard in financial research is WRDS from Wharton,
but any reputable financial data provider would work too.

[1] https://www.sciencedirect.com/journal/journal-of-econometrics/vol/187/issue/2

[2] https://quant.stackexchange.com/questions/942/any-known-bugs-with-yahoo-finance-adjusted-close-data

---

> ### Author Response · Authors · 2025-11-21
> **Addressing Comments by Reviewer SxSR (1/2)**
>
> We thank the reviewer for their detailed and constructive feedback. Below, we address each concern point-by-point.
>
> ---
>
> ### 1. Reference Integrity Issue
>
> We sincerely apologise for the confusion regarding the cited references. The reviewer is absolutely correct that the specific citations as written could not be verified in their stated venues. We want to clarify that these are real, published works and the errors stem from incorrect metadata in our citation entries.
>
> **What happened:** We used an LLM-based tool (perplexity) to bulk-convert citations from various formats (Google Scholar, inline text, notes) into BibTeX entries. This tool introduced errors through what appear to be hallucinations in the conversion process mixing and matching metadata from different sources. In some cases it used incorrect venues or journal names, in some cases the wrong year, in others wrong author names, and sometimes a combination of these issues. We failed to manually verify each converted entry against the original sources before submission. We treated this as a straightforward format conversion task and did not anticipate it could introduce substantive errors; this was an oversight on our part.
>
> We have submitted a fully corrected bibliography with the revised manuscript. To prevent similar errors, we have now manually verified all remaining references against their original sources. We deeply regret this technical error and take full responsibility for the oversight.
>
> ---
>
> ### 2. GARCH Model Selection and ARMA-GARCH Baselines
>
> We thank the reviewer for the suggestion to include additional GARCH variants, model order selection, and ARMA–GARCH models using rugarch. This is a helpful opportunity to clarify our benchmark design philosophy.
>
> #### Our Benchmark Selection Philosophy
>
> The primary goal of this paper is to assess whether continuous-time domain structure combined with neural flexibility improves forecasting performance. To this end, we include representative baselines across major modeling paradigms: econometric, neural, stochastic, and foundation models rather than exhaustive variants within any single paradigm.
>
> TimesFM is a state-of-the-art foundation model for general-purpose time-series forecasting, and serves as a substantially stronger baseline than classical econometric models in ML forecasting benchmarks. In our experiments, TimesFM already outperforms the included GARCH baselines across assets and horizons. Since UDEs in turn outperform TimesFM while using orders of magnitude fewer parameters, we did not expect higher-order or threshold-GARCH variants to materially change the comparative conclusions.
>
> #### Why We Included GARCH/FIGARCH in the First Place
>
> GARCH models are foundational in financial econometrics and also serve as standard baselines in ML time-series benchmarks. Our use of GARCH(1,1) and FIGARCH reflects this practice: they provide interpretable volatility-aware structure without neural components, positioned between purely statistical models and expressive neural architectures.
>
> #### Addressing the ARMA–GARCH Request
>
> The reviewer correctly notes ARMA-GARCH jointly models mean and volatility. Our design evaluates the same structural components and ablates the continous time-variants individually:
>
> **Component → Structure → Structure + Neural**
> - Mean reversion → ODE-MR → UDE-MR
> - Volatility clustering → FIGARCH, GARCH(1,1) → UDE-MRVC
> - Jumps → Merton → UDE-MRVCJ
>
> This enables us to quantify the marginal contribution of domain priors and highlight the central contribution of the paper:
> **UDE-MR (structure + neural correction) significantly improves over ODE-MR (structure only).**
>
> We also note that our structure-only baselines already include models capturing effects beyond ARMA–GARCH (e.g., long-memory volatility in FIGARCH and jumps in MertonJD). UDEs outperform these stronger baselines as well. We therefore do not expect ARMA–GARCH to alter our core conclusion that neural corrections to physically-motivated dynamics are key to performance improvements.

---

> > ### Author Response · Authors · 2025-11-21
> > **Addressing Comments by Reviewer SxSR (2/2)**
> >
> > #### Why We're Not Adding ARMA-GARCH or Extensive GARCH Hyperparameter Search
> >
> > In addition to the conceptual justification above, there are practical constraints.
> >
> > 1. **Computational infeasibility:** Hyperparameter search across GARCH variants would require:
> >    - a grid search over ARMA (p, q) and GARCH (m, n) orders.
> >    - Grid search over GARCH formulations (GARCH, FIGARCH)
> >    - Across 10 assets × 30+ years of daily data × multiple forecast horizons × rolling window validation
> >    This represents hundreds of model configurations and would require weeks of computation time, far beyond the scope of a two-week revision.
> >
> > 2. **Technical barriers:** Incorporating a separate R toolchain solely for ARMA–GARCH would introduce engineering complexity and reduce experimental consistency, particularly given the tightly integrated Python-based infrastructure (neuromancer + pytorch + numpy) for Neural ODE training and rolling-window evaluation.
> >
> > More broadly, expanding only the econometric family would create benchmark asymmetry: by the same logic, one could request additional stochastic volatility models (e.g., SABR) or a wider spectrum of state-space neural architectures. Our benchmark suite instead includes representative models across econometric, neural, stochastic, and foundation paradigms, which aligns with the methodological focus of this work.
> >
> > We therefore believe the current baseline set adequately supports the scientific conclusions. That said, we are happy to consider ARMA–GARCH extensions in future iterations if reviewers view this as important for the final version.
> >
> > ---
> >
> > ### 3. Yahoo Finance Data Quality
> >
> > We thank the reviewer for raising this point and agree that our language around data quality should be revised. While Yahoo Finance is widely used in academic machine learning research due to its accessibility and convenience, it is not considered a high-quality data source relative to institutional providers such as Bloomberg or WRDS. These institutional data sources require paid subscriptions or academic partnerships and are not broadly accessible for reproducible ML research, especially outside production finance environments. We have updated the manuscript to reflect this distinction clearly.
> >
> > To clarify our usage context:
> > - Our dataset consists solely of major financial indices (S&P 500, FTSE 100, DAX, Nikkei 225, Hang Seng) and cryptocurrencies (BTC, ETH).
> > - Index-level prices do not suffer from corporate action adjustment issues that primarily affect individual equities.
> > - The main reliability concerns reported around Yahoo Finance tend to relate to unadjusted corporate actions, which do not affect the index-level data used in this paper.
> >
> > In the revision, we have:
> > - removed any phrasing implying Yahoo Finance is “high-quality,” and instead describe it as publicly accessible and commonly used in ML benchmarking, and
> > - explicitly note that institutional-grade data would be required for deployment in real-world trading or risk management systems.
> >
> > We believe this clarification addresses the reviewer’s concern while keeping the focus on the paper’s contribution.

---

> > > ### Comment · Reviewer_SxSR · 2025-11-24
> > > **Reply to Comments**
> > >
> > > Thanks for the detailed reply regarding my previous review comments.
> > >
> > > 1. I can confirm that the references are updated in the most up-to-date version, and I appreciate the authors' honesty and quick fixes. I do not have any further comment regarding this point. I understand the usage of AI/LLM tools for academic writing, but the consequences of such usage is probably not yet clear and such conversation is beyond the scope of the review of this article. Specifically for this review, I am not sure what is the policy at TMLR regarding LLM usage for papers, and I believe this is the decision of the editor.
> > >
> > > 2.  The authors' response claim to have edited the paper regarding the potential limitation of using YF data and removed the language on YF being "high-quality," but I can't see the changes made on this. I wonder if the authors could double-check on this. (Line 25, Page 1 is still the same as in the last version.)
> > >
> > > 3. I understand the concern regarding the computational time on the "hyper-parameter search" for the GARCH family of models, but I still think it's valuable to see at least one result on ARMA-GARCH. The easiest would be ARMA(1,1)-GARCH(1,1), and this should work as a good representative. I know there are packages to call R packages/functions from Python, and that might help with the pipelines.

---

> > > > ### Author Response · Authors · 2025-11-25
> > > > **Reply to Comments**
> > > >
> > > > 2. Yahoo Finance wording
> > > > Thanks for pointing out. The “high-quality” phrasing  has now been fully corrected in the latest compiled version.
> > > >
> > > > 3. ARMA–GARCH baseline
> > > > We agree that an ARMA(1,1)–GARCH(1,1) baseline would strengthen the paper. Before proceeding, we would like to clarify whether you view this as essential for the paper's core scientific contribution, or as a valuable but non-critical addition. We ask because the rolling-window evaluation involves non-trivial cloud compute costs.
> > > >
> > > > If this baseline is deemed necessary, we plan to implement it using a Python package (armagarch) to maintain consistency with our existing pipeline. Integrating R-based tools like rugarch would require managing cross-language dependencies, data marshaling, and reproducibility across environments , engineering overhead we'd prefer to avoid unless you believe the implementation differences are substantial enough to justify it.
> > > >
> > > > Please confirm: (1) whether you consider the ARMA–GARCH baseline truly essential, and (2) whether a Python implementation would be acceptable. If essential, we're happy to proceed and would request a brief extension to properly incorporate these experiments.

---

> > > > > ### Comment · Reviewer_SxSR · 2025-11-25
> > > > > **Further comments**
> > > > >
> > > > > 1. I think ARMA-GARCH would be a valuable inclusion as a comparison, but if it involves significant cost for doing so, I doubt those experiments will change much to the results.
> > > > >
> > > > > 2. I can confirm that the YF sentence is revised.
> > > > >
> > > > > 3. Just noticed the following typo on page 11, line 302, a comma is placed after the space, i.e. "...regimes ,an essential..."
> > > > >
> > > > > I don't have further comments on this paper. Thanks to the authors for the work and efforts to address my comments.

---

### Decision · Action_Editor_YQ6J · 2025-12-21

**Recommendation:** Accept with minor revision

**Additional Comments:**

Based on the provided reviews and the subsequent author-reviewer interactions, the paper has been significantly improved. The authors' successful revisions led to a consensus among the reviewers; thus, I recommend the paper for acceptance given that the following points can be ensured:

1. A few remaining typos (e.g., misplaced commas in lines 108 and 302) should be corrected.
2. The authors must ensure the bibliography is completely clean, as an earlier version contained hallucinated metadata from an LLM-based tool. Moreover, the authors should carefully check any part of the paper where LLMs had been used. As per TMLR policy, “Whichever tools are used, authors are fully responsible for content on which they are listed as (co-) authors. This includes, but is not limited to, content generated by LLMs that could be construed as plagiarism or scientific misconduct (e.g., fabrication of facts).”

**Audience:**

Yes

**Audience Explanation:**

The reviewers agree that the work is relevant to individuals in TMLR's audience that are interested in topics at the intersection of scientific machine learning and financial applications.

**Claims And Evidence:**

Yes

**Claims Explanation:**

While initial reviews raised concerns regarding presentation (treatment of jump discontinuities; issues in references), empirical evaluation (metrics, datasets), and baselines (performance of random walk & GARCH), the authors successfully addressed these in their revisions.

---

> ### Author Response · Authors · 2025-12-24
> **Camera-Ready Revisions Completed**
>
> Dear Action Editor,
>
> Thank you for the recommendation. We appreciate the thorough evaluation by the reviewers and your guidance in improving the manuscript. We have completed the requested minor revisions:
> Typos: We have corrected all remaining typos, including the misplaced commas in lines 108 and 302.
> Bibliography and LLM usage: We have thoroughly reviewed and verified the bibliography to ensure all entries are accurate and free from any LLM-generated errors. Additionally, we confirm that no LLMs were used in any other sections of the paper.
>
> The manuscript has also been de-anonymized, and line numbers have been removed as required for the camera-ready version.
>
> Best Regards